# Improving self-training under distribution shifts via anchored confidence with theoretical guarantees

**Taejong Joo & Diego Klabjan**
Department of Industrial Engineering & Management Sciences
Northwestern University
Evanston, IL, USA
{taejong.joo,d-klabjan}@northwestern.edu

## Abstract

Self-training often falls short under distribution shifts due to an increased discrepancy between prediction confidence and actual accuracy. This typically necessitates computationally demanding methods such as neighborhood or ensemble-based label corrections. Drawing inspiration from insights on early learning regularization, we develop a principled method to improve self-training under distribution shifts based on temporal consistency. Specifically, we build an uncertainty-aware temporal ensemble with a simple relative thresholding. Then, this ensemble smooths noisy pseudo labels to promote selective temporal consistency. We show that our temporal ensemble is asymptotically correct and our label smoothing technique can reduce the optimality gap of self-training. Our extensive experiments validate that our approach consistently improves self-training performances by 8% to 16% across diverse distribution shift scenarios without a computational overhead. Besides, our method exhibits attractive properties, such as improved calibration performance and robustness to different hyperparameter choices.

## 1 Introduction

In this work, we address the challenge of adapting pre-trained neural networks at test time under distribution shifts, a problem known as test-time adaptation (TTA) or source-free domain adaptation (SFDA). Distribution shifts—where a model trained on one distribution is then tested on a different one—are ubiquitous in many practical scenarios due to demographic subpopulation shift [1] and changes in data collection environments [2–4]. Despite the robust performance of neural networks under independent and identically distributed (i.i.d.) settings, they often suffer from substantial performance degradation under such shifts [5, 6]. Recently, TTA and SFDA have proven their effectiveness in resolving these critical issues by effectively leveraging information about the distribution shifts contained in unlabeled samples given at the test time.

Self-training is the basis of many state-of-the-art methods in TTA and SFDA [7–9], utilizing pseudo labels generated from the model's own predictions to train a model on unlabeled samples [10]. Since pseudo labels are regarded as true labels in self-training, the success of self-training methods highly depends on how to filter incorrect pseudo labels to prevent self-confirmation bias [11]. This issue has been effectively handled via simple confidence-based thresholding in i.i.d. settings [12–14]. However, the distribution shifts make it hard to filter incorrect pseudo labels due to high noise rates even under high threshold [15]. Thus, sophisticated methods filter incorrect pseudo labels based on a neighborhood structure of the data [16–18] and consistency of multiple predictions under different models [19] or augmentations [20](cf. Section 5), which are computationally intensive by nature.

Recent insights in [8] suggest an alternative strategy can be also effective: promoting temporal consistency can enhance self-training performance in SFDA without the computational burden of

38th Conference on Neural Information Processing Systems (NeurIPS 2024).

previous methods. The temporal consistency regularizer, so called early learning regularization (ELR) [21], was originally developed to address neural networks' tendency to learn clean information first and then gradually memorize noisy labels [22, 23]; this setting is naturally connected to self-training scenarios when we regard the pseudo labels as random noisy labels. However, the impacts of ELR on self-training have not been fully understood. Also, since ELR does not consider unique characteristics of distribution shifts, we aim to answer the following question: *Is there any principled way to improve the way of memorizing all past predictions tailored for self-training under distribution shifts?*

In this work, we show that the answer is affirmative by proposing Anchored Confidence (AnCon) that uses *confident predictions* to support a generalized notion of *temporal consistency*. Specifically, we construct a generalized temporal ensemble, which weighs predictions based on predictive uncertainty, and then use the ensemble as a smoothing vector in label smoothing [24]. Then, through rigorous theoretical analyses, we show that our simple heuristic for the generalized temporal ensemble is asymptotically correct and that the label smoothing formulation can reduce the optimality gap. As a result, AnCon can correct wrong pseudo labels without expensive computations and can be easily applied to self-training methods by replacing one-hot pseudo labels with smooth pseudo labels, unlike neighborhood-based or centroid methods. Through extensive experiments, we show that AnCon improves self-training under diverse distribution shift scenarios and posses many attractive properties.

Our contribution can be summarized as follows: **1)** We develop AnCon, which is the first algorithm that attempts to improve self-training under distribution shifts by generalizing a notion of temporal consistency with theoretical guarantees; **2)** Without any additional forward passes or neighborhood search, AnCon improves self-training performances 8% and 16% under domain shifts and image corruptions, respectively; **3)** Remarkably, we also show that AnCon significantly improves calibration performance and is robust with respect to model selection methods and hyperparameter choices.

## 2   Background

**Notation and setup** For an input space $\mathcal{X} \subseteq \mathbb{R}^d$ and a label space $\mathcal{Y} = [K] := \{1, 2, \cdots, K\}$, we define $X$ and $Y$ to be random variables of input and output with probability densities $p_X$ and $p_Y$, respectively. We also define a neural network $f(\cdot; \theta) : \mathcal{X} \to \triangle^{K-1}$ parameterized by a parameter $\theta \in \mathbb{R}^p$ where $\triangle^{K-1}$ is the probability simplex with $K$ elements. Our goal is to minimize the cross-entropy loss $\min_{\theta \in \mathbb{R}^p} l(\theta) := \mathbb{E}_{XY}[H(f(X; \theta), p_{Y|X})]$ where $H(f(x; \theta), p_{Y|X=x}) := -\sum_{k \in [K]} p(Y = k|x) \log f_k(x; \theta)$. In the SFDA setting, we are given an initial parameter $\theta_0$ that is trained on a different data generating distribution $(X', Y')$, e.g., $\theta_0 \in \arg\min_{\theta \in \mathbb{R}^p} \mathbb{E}_{X'Y'}[H(f(X'; \theta), p_{Y'|X'})]$. We can think about this setting as either (1) $(X', Y')$ being pre-training data and $\theta_0$ the foundation model with the task of fine-tuning the model on unlabeled $X$ or (2) a transfer learning problem with $(X', Y')$ being the source domain data and $X$ the target domain. Here, we assume only covariate shift without concept shift; that is, $p_X \neq^D p_{X'}$ but $p_{Y|X} =^D p_{Y'|X'}$. Even in this case, we note that suboptimality under the distribution shift, i.e., $\min_{\theta \in \mathbb{R}^p} l(\theta) - l(\theta_0)$, can be large.

**Self-training** In this work, we tackle the distribution shifts by using the self-training method that replaces the true label $Y(x)$ by the pseudo label, i.e., $\min_{\theta_{new}} \hat{l}(\theta_{new}; \theta) = \mathbb{E}_X[-\log f_{\hat{Y}(x;\theta)}(x; \theta_{new})]$ where $\hat{Y}(x; \theta) := \arg\max_{k \in [K]} f_k(x; \theta)$ is a pseudo label under $\theta$. Specifically, the algorithmic framework of self-training is as follows: given $\theta_0$, we iteratively find model parameter $\theta_{m+1}$ for $m = 0, 1, \cdots$ with $\theta_{m+1} \in \arg\min_{\theta_{new}} \hat{l}(\theta_{new}; \theta_m)$. For later use, we also define a prediction confidence $c(x; \theta) := \max_{k \in [K]} f_k(x; \theta)$ and $\theta_{0:m} := (\theta_0, \cdots, \theta_m)$.

**Early learning regularization** In the learning from noisy labels (LFN) scenario, [21] identified an "early-learning phenomenon" where neural networks initially learn information contained in clean labels before gradually memorizing noisy labels, leading to a performance deterioration as training progresses. To mitigate this issue, ELR penalizes predictions that deviate from earlier predictions by defining a target network from the past predictions: $\bar{f}_{ELR}(x; \theta_{0:m}) := \sum_{j=0}^{m}(1 - \gamma) \cdot \gamma^{m-j} f(x; \theta_j)$. Then, adding an auxiliary loss of $L_{ELR}(\theta; \theta_{0:m}) = \mathbb{E}_X[\log(1 - f(X; \theta)^T \bar{f}_{ELR}(X; \theta_{0:m}))]$ to $\hat{l}(\theta; \theta_m)$ can prevent memorization of noisy labels while preserving correct patterns.

Notably, this insight has recently been confirmed to be applicable in the SFDA setting by [8]. This observation is appealing because ELR can be efficiently implemented by reusing past predictions without additional forward passes or neighborhood searching, unlike dominant methods in SFDA

[16–20]. Nevertheless, given that ELR stems from a general property of the neural network training in the i.i.d. setting, herein we aim to step towards a more principled approach to encourage temporal consistency tailored for distribution shift scenarios.

## 3 Anchored confidence

In this section, we introduce `AnCon`, which promotes the temporal consistency on selectively chosen predictions via label smoothing. In Section 3.1, we first explain the idea of promoting selective temporal consistency based on confident predictions via label smoothing [24] with a temporal ensemble. Then, in Section 3.2, we explain how to effectively construct temporal ensemble for improving self-training under distribution shifts. Finally, we theoretically analyze the efficacy of `AnCon` by drawing connection between our method and knowledge distillation in Section 3.3.

### 3.1 Selective temporal consistency via label smoothing

In this work, we utilize label smoothing [24] to promote the selective temporal consistency instead of using an auxiliary loss function like ELR. Specifically, given a generalized temporal ensemble $\bar{f}(x; \theta_{0:m}, \mathbf{w}_{0:m})$ with $\mathbf{w}_{0:m} := (w_0, \cdots, w_m)$ which will be specified in Section 3.2, we construct a regularized pseudo label $\tilde{Y}(X; \theta_{0:m}, \mathbf{w}_{0:m})$ by using $\bar{f}(x; \theta_{0:m}, \mathbf{w}_{0:m})$ as a smoothing vector for the pseudo label $\hat{Y}(x; \theta_m)$. That is, we perform self-training by

$$\min_\theta \mathbb{E}_X[H(f(X; \theta), \tilde{Y}(X; \theta_{0:m}, \mathbf{w}_{0:m}))], \quad \tilde{Y}(X; \theta_{0:m}, \mathbf{w}_{0:m}) = (1-\lambda)E_1(\hat{Y}(x; \theta_m)) + \lambda\bar{f}(x; \theta_{0:m}, \mathbf{w}_{0:m})$$

(1)

where $\lambda \in [0, 1]$ is a coefficient, $E_1(\cdot)$ is the one-hot encoding, and $\bar{f}(x) := (\bar{f}_1(x), \cdots, \bar{f}_K(x))$ is the $K$-dimensional output of the generalized temporal ensemble (cf. (2)).

Thus, `AnCon` can control the usage of potentially noisy information in $\hat{Y}(x; \theta_m)$ based on its consistency with $\bar{f}(x; \theta_{0:m}, \mathbf{w}_{0:m})$. Not only can this approach preserve the early learning phenomenon as in ELR (cf. Section 2), but the label smoothing formulation also significantly stabilizes the self-training performance under different hyperparameter choices due to the fact that the optimal values of hyperparameters are less problem dependent compared to its equivalent auxiliary regularization [25]. Beyond removing the burden of hyperparameter search, this robustness is a particularly intriguing property under distribution shift scenarios where the model selection becomes a challenging task.

Further, encouraging the temporal consistency through label smoothing enables us to connect our method with knowledge distillation (KD) [26], which can provide a wide range of principled techniques and theoretical results developed in KD. As a concrete example, we will show when and how `AnCon` can reduce the optimality gap of self-training, i.e., a case with $\lambda = 0$, in Section 3.3.

### 3.2 Constructing an effective generalized temporal ensemble with prediction confidences

Next, we construct the generalized temporal ensemble that makes the selective temporal consistency in (1) work effectively in self-training under distribution shifts. Specifically, given $\theta_{0:m}$ and weights $\mathbf{w}_{0:m}(x) \in \triangle^m$ for each $x \in \mathcal{X}$, the prediction by the generalized temporal ensemble is

$$\bar{f}_k(x; \theta_{0:m}, \mathbf{w}_{0:m}) := \sum_{i=0}^m w_i(x) \cdot p(y = k | x, \theta_i), \quad k \in [K]$$

(2)

where $p(y|x, \theta_i)$ is the prediction made by $f(x; \theta_i)$, which can be either soft ($p(y = j|x, \theta_i) = f_j(x; \theta_i)$) or hard ($p(y = j|x, \theta_i) = 1$ if $j = \arg\max_{k \in [K]} f_k(x; \theta_i)$ and $p(y = j|x, \theta_i) = 0$ otherwise). In this work, we use hard prediction because soft prediction puts more weights on recent predictions since self-training tends to keep increasing the prediction confidence during training.

Surprisingly, we will show that the following simple relative thresholding for determining $\mathbf{w}_{0:m}$ gives the asymptotic optimal weights achieving the minimum worst-case optimality gap of self-training:

$$w_m(x) \propto \mathbf{1}(c(x; \theta_m) > \delta_m^{(\beta)}), \quad \delta_m^{(\beta)} := \sum_{i=0}^m (1 - \beta)\beta^{m-i}\hat{\mathbb{E}}_X[c(X; \theta_i)]$$

(3)

where $\delta_m^{(\beta)}$ is an exponential moving average (EMA) of prediction confidence with hyperparameter $\beta$ and $\hat{\mathbb{E}}[c(X; \theta_i)]$ is a Monte-Carlo approximation of $\mathbb{E}[c(X; \theta_i)]$ with mini-batch samples.

Intuitively, our weighting mechanism aggregates only relatively confident predictions with a uniform weight. Given the observation that relative ordering of confidence is highly correlated with accuracy even under distribution shifts [27, 28], our thresholding rule would tend to put non-zero weights on correct predictions. Besides, by employing the relative criterion, the thresholding does not suffer from the problems that neglect predictions obtained in the early stage of training.

We also remark that `AnCon` has almost the same computational cost as ELR. Specifically, (2) can be implemented by $\bar{f}_k(x; \theta_{0:m}, \mathbf{w}_{0:m}) = \bar{f}_k(x; \theta_{0:m-1}, \mathbf{w}_{0:m-1}) + w_m(x)p(y = k|x, \theta_m)$ that requires to store the weighted sum of previous predictions without additional forward passes or storing previous parameters, which is the same as storing the previous logit vector in ELR. Similarly, (3) can be efficiently implemented by $\delta_m^{(\beta)} = \beta \delta_{m-1}^{(\beta)} + (1-\beta)\hat{\mathbb{E}}_X[c(X; \theta_m)]$ that requires constant additional computational costs compared to vanilla self-training with the constant being small. Therefore, `AnCon` shares the same computational benefits as ELR compared to other state-of-the-art methods in SFDA.

**On optimality of the relative thresholding in (3)** From the optimization perspective, the optimal weights $\mathbf{w}_{0:m}^{\dagger}$ correspond to the weights under which self-training with $\tilde{Y}(X; \theta_{0:m}, \mathbf{w}_{0:m})$ can minimize the expected loss:

$$\mathbf{w}_{0:m}^{\dagger} \in \arg\min_{\mathbf{w}_{0:m}} l(\theta_{\mathbf{w}_{0:m}}^{\dagger}), \quad \theta_{\mathbf{w}_{0:m}}^{\dagger} \in \arg\min_{\theta} \hat{\mathbb{E}}_X[H(f(X;\theta), \tilde{Y}(X; \theta_{0:m}, \mathbf{w}_{0:m}))]. \quad (4)$$

Unfortunately, $l(\theta_{\mathbf{w}_{0:m}}^{\dagger})$, or its empirical counterpart, is not available in self-training due to the absence of labels. Further, even if labels are given, solving (4) is intractable due to non-smoothness of $l(\theta_{\mathbf{w}_{0:m}}^{\dagger})$ with respect to $\mathbf{w}_{0:m}$ and the cost of finding $\theta_{\mathbf{w}_{0:m}}^{\dagger}$.

To circumvent this issue, we show in Section 3.3 that (4) can be relaxed to the problem of finding ensemble weights that give a maximum likelihood estimation (MLE) solution under certain conditions. As a result, instead of solving the intractable optimization in (4), we find the optimal weights by

$$\mathbf{w}_{0:m}^{\dagger} \in \arg\max_{\mathbf{w}_{0:m}} \mathbb{E}_{XY}[\log \bar{f}_Y(X; \theta_{0:m}, \mathbf{w}_{0:m})]. \quad (5)$$

In the following theorem which is proven in Appendix B.2, we show that the simple relative thresholding in (3) can make $\bar{f}(x; \theta_{0:m}, \mathbf{w}_{0:m})$ asymptotically correct for samples where the neural network tends to be relatively confident during self-training and thus our simple weighting mechanism in (3) to be the solution of (5) in the asymptotic region.

**Theorem 3.1.** *Let $A_i(c) := \{x \in \mathcal{X} | c(x; \theta_i) > c\}$, $Q(x; \mathbf{c}_{0:m}) := \sum_{i=0}^m \mathbf{1}(x \in A_i(c_i))$, and $\bar{p}(x; \mathbf{c}_{0:m}) = \frac{1}{Q(x; \mathbf{c}_{0:m})} \sum_{i=0}^m \mathbb{E}_{Y|X=x}[\mathbf{1}(Y(x) = \hat{Y}(x; \theta_i))]\mathbf{1}(x \in A_i(c_i))$ for $x$ such that $Q(x; \mathbf{c}_{0:m}) > 0$. Let us assume that random events $\mathbf{1}(Y(x) = \hat{Y}(x; \theta_i))$ and $\mathbf{1}(Y(x) = \hat{Y}(x; \theta_j))$ are conditionally independent given $X \in A_i(c)$ for $j \in \{0, \cdots, i-1\}$, $x \in \mathcal{X}$, $c \in [0, 1)$. If $x \in \mathcal{X}$ such that $\bar{p}(x; \mathbf{c}_{0:m}) > 1/2$, then for the generalized temporal ensemble in (3), it holds that*

$$p(\arg\max_{k \in [K]} \bar{f}_k(x; \theta_{0:m}, \mathbf{w}_{0:m}) \neq Y(x)) \leq \exp\left(-\frac{Q(x; \mathbf{c}_{0:m})}{2} \cdot \xi(\bar{p}(x; \mathbf{c}_{0:m}))\right) \quad (6)$$

*where $\xi(z) := 2z - 1 - \log(2z)$ is a positive increasing function in $z \in [0.5, 1]$.*

The result states that as long as the average accuracy for *relatively confident predictions* over iterations exceeds 50%, the error rate of the generalized temporal ensemble monotonically decreases as $Q(x; \mathbf{c}_{0:m})$ increases. Furthermore, $\bar{f}(x; \theta_{0:m}, \mathbf{w}_{0:m})$ is asymptotically correct on $x$ such that $Q(x; \mathbf{c}_{0:m}) \to \infty$ as $m \to \infty$. `AnCon` aims to achieve these desirable properties through the *uncertainty-aware* temporal consistency that helps to satisfy the condition $\bar{p}(x; \mathbf{c}_{0:m}) > 0.5$. Specifically, as shown in Figure 4a in Appendix, our generalized temporal ensemble's accuracy tends to significantly increase as the number of confident samples increases, being consistent with our theory. We note that this monotonic improvement would not be the case for the temporal ensemble without uncertainty-awareness and vanilla self-training (cf. Figure 4a).

Finally, we emphasize that the assumption $\bar{p}(x; \mathbf{c}_{0:m}) > 0.5$ applies only to *relatively confident predictions* which are *averaged over iterations*. This is significantly weaker than requiring a lower bound of an expected accuracy of *each sample* for *every iteration*, which is the case when LFN methods are directly applied to the self-training scenario. Also, due to its dependency on the choice of the confidence thresholds $\mathbf{c}_{0:m}$, the assumption can hold by controlling $\mathbf{c}_{0:m}$ at the expense

of loosening the upper bound in (6) (e.g., selecting 90th-quantile as in Figure 4b in Appendix). Specifically, increasing the thresholds can improve $\xi(\bar{p}(x; \mathbf{c}_{0:m}))$ and enhance the chance of satisfying $\bar{p}(x; \mathbf{c}_{0:m}) > 1/2$ but reducing $Q(x; \mathbf{c}_{0:m})$. While this trade-off necessitates a proper choice of $\mathbf{c}_{0:m}$, our extensive experiments show that setting the threshold $c_m$ by the EMA of prediction confidence, i.e., $\delta_m^{(\beta)}$ in (3), works effectively.

### 3.3 Theoretical insights from knowledge distillation

In this section, we present a novel connection between AnCon and KD for addressing intractability of (4). KD is a framework for training a small student network $f$, e.g., ResNet-50 [29], with an additional supervision from a large teacher network $f^{(t)}$, e.g., ResNet-152. Specifically, a cross-entropy under KD is $l_{KD}(\theta) = \mathbb{E}_{XY}\left[H(f(X; \theta), (1 - \lambda_{KD})E_1(Y(X)) + \lambda_{KD}f^{(t)}(X))\right]$ with $\lambda_{KD} \in [0, 1]$, which bears a significant similarity with AnCon (cf. (1)). Indeed, AnCon can be understood as a special case of KD called self-distillation when $f$ and $f^{(t)}$ have the same architecture, where the generalized temporal ensemble $\bar{f}$ corresponds to the teacher network $f^{(t)}$ with a notable difference that the pseudo label $\hat{Y}$ is used instead of the true label $Y$. Based on this connection, we perform a convergence analysis of AnCon by modifying the partial variance reduction theory [30], as given below. We note that the usage of $\tilde{Y}$ and $\bar{f}$ results in an inherently biased gradient estimator, which requires special treatments for the convergence analysis unlike the typical self-distillation setting.

**Setup** Following [30], we assume a linear model $f_k(x; \theta) = \exp(\Theta_k^T x)/\sum_{i \in [K]} \exp(\Theta_i^T x)$ with $\Theta_i \in \mathbb{R}^d$ for $i \in [K]$, $\theta := \text{Concat}(\Theta_1, \cdots, \Theta_K) \in \mathbb{R}^{dK}$, and $\text{Concat}(\cdot)$ is the concatenation operation. Also, we assume a bounded support for $X$; that is, $\| x \| \leq C$ for all $x$ where $p_X(x) > 0$. Under this setting, we repeat the following steps starting from a given $\theta_0$ (i.e., $m = 0$):

1. Outer temporal ensemble update: Update $\mathbf{w}_{0:m}$ by (3) to obtain $\bar{f}(x; \theta_{0:m}, \mathbf{w}_{0:m})$ (cf. (2)).
2. Inner parameter update: With $\theta_{m,0} := \theta_m$ and $(\theta_{0:m}, \mathbf{w}_{0:m})$, solve (1) with stochastic gradient descent $\theta_{m,t+1} = \theta_{m,t} - \gamma g_\xi^{(m,t)}$ for $t \in \{0, 1, \cdots, T - 1\}$ and set $\theta_{m+1} = \theta_{m,T}$.
3. Iteration number update: Set $m = m + 1$ and terminate if $m = \hat{T}$.

Here, $\gamma$ is the learning rate, $\xi$ contains $b$ random samples from $p_X$, and the stochastic gradient under AnCon is defined as $g_\xi^{(m,t)} := \nabla_{\theta_{m,t}} \frac{1}{b}\sum_{X_i \in \xi} H(f(X_i; \theta_{m,t}), \tilde{Y}(X_i; \theta_{0:m}, \mathbf{w}_{0:m}))$. For the linear model, we note that $g_\xi^{(m,t)} = \nabla \tilde{l}_\xi(\theta_{m,t}) - \lambda \hat{g}_\xi(\theta_{0:m}, \mathbf{w}_{0:m})$ where $\tilde{l}_\xi(\theta_{m,t}) = \frac{1}{b}\sum_{X_i \in \xi} H(f(X_i; \theta_{m,t}), \hat{Y}(X_i; \theta_m))$ and $\hat{g}_\xi(\theta_{0:m}, \mathbf{w}_{0:m}) := \text{Concat}(\hat{g}_{1,\xi}(\theta_{0:m}, \mathbf{w}_{0:m}), \cdots, \hat{g}_{K,\xi}(\theta_{0:m}, \mathbf{w}_{0:m}))$ with $\hat{g}_{k,\xi}(\theta_{0:m}, \mathbf{w}_{0:m}) := \frac{1}{b}\sum_{X_i \in \xi}[(\bar{f}_k(X_i; \theta_{0:m}, \mathbf{w}_{0:m}) - \hat{Y}_k(X_i; \theta_m))X_i]$ for $k \in [K]$.

In Theorem 3.2 which is proven in Appendix B.3, we analyze the convergence of the inner parameter update step under AnCon and vanilla self-training.

**Theorem 3.2.** *Let us assume $l(\theta)$ satisfies L-smoothness, $\mathcal{L}$-expected smoothness, and $\mu$-Polyak-Lojasiewicz (PL) condition (cf. Assumptions B.1, B.2, and B.3 in Appendix B.1). For $\gamma \leq \frac{\mu}{4\mathcal{L} \cdot L}$, a carefully chosen $\lambda$ (cf. $\lambda = \lambda_m^\dagger := \frac{\mathbb{E}_\xi[\langle \nabla \tilde{l}_\xi(\theta^*), \hat{g}_\xi(\theta_{0:m}, \mathbf{w}_{0:m})\rangle]}{\mathbb{E}_\xi\|\hat{g}_\xi(\theta_{0:m}, \mathbf{w}_{0:m})\|^2 + \frac{2}{L\gamma}\|\hat{g}(\theta_{0:m}, \mathbf{w}_{0:m})\|^2}$ in Lemma B.6 in Appendix B.5), and any realization of $\theta_m$, it holds that*

$$\underbrace{\mathbb{E}[l(\theta_{m,t}) - l(\theta^*)|\theta_m]}_{\text{Optimality gap}} \leq \underbrace{(1 - \gamma\mu)^t(l(\theta_m) - l(\theta^*))}_{\text{Improvement over the initial model}} + \underbrace{\frac{8C^2}{\mu}g^\mathcal{E}(\theta_m)}_{\text{Bias of the pseudo label}} + \underbrace{\frac{2}{\mu}N(\lambda_m^\dagger; \theta_{0:m}, \mathbf{w}_{0:m})}_{\text{Neighborhood size}}$$

(7)

*where $l^* := l(\theta^*)$ with $\theta^* \in \arg\min_{\theta \in \Theta} l(\theta)$, $g^\mathcal{E}(\theta) := \mathbb{E}[\mathbf{1}(\hat{Y}(X; \theta) \neq Y(X))]$, $N(\lambda; \theta_{0:m}, \mathbf{w}_{0:m}) = \lambda^2 \| \hat{g}(\theta_{0:m}, \mathbf{w}_{0:m}) \|^2 + \frac{L\gamma}{2}\mathbb{E}_\xi\left[\| \nabla \tilde{l}_\xi(\theta^*) - \lambda \hat{g}_\xi(\theta_{0:m}, \mathbf{w}_{0:m}) \|^2\right]$, and $N(\lambda_m^\dagger; \theta_{0:m}, \mathbf{w}_{0:m}) \leq N(0)$ where $N(0) := N(0; \theta_{0:m}, \mathbf{w}_{0:m})$ for any $(\theta_{0:m}, \mathbf{w}_{0:m})$.*

*Further, when $\mathbf{w}_{0:m}$ is such that $\mathbb{E}_\xi[\langle \nabla \tilde{l}_\xi(\theta^*), \hat{g}_\xi(\theta_{0:m}, \mathbf{w}_{0:m})\rangle] \geq 0$, i.e., the teacher has a sufficiently good performance, it holds that*

$$\frac{N(\lambda_m^\dagger; \theta_{0:m}, \mathbf{w}_{0:m})}{N(0)} \leq \min\left(1, 2\min\left(1, \frac{C^2 g^{KL}(\theta_{0:m}, \mathbf{w}_{0:m})}{\sigma_* \sigma_{(\theta_{0:m}, \mathbf{w}_{0:m})}}\right) + \frac{2C^2 g^C(\theta_{0:m}, \mathbf{w}_{0:m})}{L\gamma \sigma^2_{(\theta_{0:m}, \mathbf{w}_{0:m})}}\right)$$

(8)

*where* $\sigma_*^2 := \mathbb{E}_\xi \parallel \nabla \tilde{l}_\xi(\theta^*) \parallel^2$, $\sigma_{(\theta_{0:m}, \mathbf{w}_{0:m})}^2 := \mathbb{E}_\xi \parallel \hat{g}_\xi(\theta_{0:m}, \mathbf{w}_{0:m}) \parallel^2$, $g^{KL}(\theta_{0:m}, \mathbf{w}_{0:m}) = \mathbb{E}_X[D_{KL}(f(X; \theta^*) \parallel \bar{f}(X; \theta_{0:m}, \mathbf{w}_{0:m}))]$ *with* $D_{KL}(p \parallel q)$ *is the KL-divergence between* $p$ *and* $q$, *and* $g^C(\theta_{0:m}, \mathbf{w}_{0:m}) = \parallel \mathbb{E}_X[\bar{f}(X; \theta_{0:m}, \mathbf{w}_{0:m})] - \mathbb{E}_X[\hat{Y}(X; \theta_m)] \parallel^2$.

In Theorem 3.2, (7) characterizes the optimality gap in the inner loop optimization. Specifically, the first term is about reducing the initial optimality gap over iterations and motivates why we need *adaptation*, e.g., by self-training, if the performance deteriorates under severe distribution shifts. The second term is about the bias of the pseudo label and motivates the challenges of self-training under poorly performing pseudo labels in the case of severe distribution shifts. Crucially, these two terms can be fully characterized by the quality of initial model $\theta_m$ and do not depend on $\mathbf{w}_{0:m}$. Thus, we concentrate on the impacts of $\mathbf{w}_{0:m}$ designed in (1)-(3) on $N(\lambda_m^\dagger; \theta_{0:m}, \mathbf{w}_{0:m})$ to show that `AnCon`'s effectiveness on improving self-training performance under distribution shifts.

First, Theorem 3.2 shows the effectiveness of our label smoothing formulation in (1) under properly chosen $\lambda_m^\dagger$. Specifically, compared to vanilla self-training, `AnCon` results in the smaller neighborhood size of the stochastic gradient descent; $N(\lambda_m^\dagger; \theta_{0:m}, \mathbf{w}_{0:m}) \leq N(0)$. That is, the result suggests that `AnCon` is at least better than vanilla self-training under the mild regularity conditions.

Further, under the additional assumption of a sufficiently good performance temporal ensemble, (8) motivates `AnCon`'s weighting mechanism as a relaxed solution of the intractable optimization problem in (4). Specifically, if the marginal distribution of the pseudo labels does not change quickly over outer iterations (which is the case especially for the later training stages as shown in Figure 5 in Appendix), changing $\mathbf{w}_{0:m}$ would have only a marginal impact on $g^C(\theta_{0:m}, \mathbf{w}_{0:m})$. Thus, the weighting mechanism that minimizes $g^{KL}(\theta_{0:m}, \mathbf{w}_{0:m})$ would minimize the worst-case optimality gap, which justifies our approach of circumventing intractability of (4) with (5). In this regard, `AnCon`'s weighting mechanism in (3) could be thought of as a relaxed solution of (4) as it minimizes $g^{KL}(\theta_{0:m}, \mathbf{w}_{0:m})$ in the asymptotic region (cf. Theorem 3.1).

We conclude this section by analyzing the three iterative steps where the pseudo labels and the temporal ensembles keep updating.

**Corollary 3.2.1.** *Let us assume* $l(\theta)$ *satisfies L-smoothness, $\mathcal{L}$-expected smoothness, and $\mu$-PL condition. For $\gamma \leq \frac{\mu}{4\mathcal{L} \cdot L}$, $\lambda$ carefully adjusted at each outer temporal ensemble update (cf. $\lambda = \lambda_j^\dagger$ in Lemma B.6 for each outer iteration $j \in \{0, \cdots, \hat{T} - 1\}$), and any initial parameter $\theta_0$, it holds that*

$$\mathbb{E}[l(\theta_{\hat{T}}) - l^*] \leq (1 - \mu\gamma)^{T \cdot (\hat{T}-1)}(l(\theta_0) - l^*) + \zeta_{\hat{T}} \mathbb{E}_{j \sim I^{(\hat{T})}}\left[\frac{8C^2}{\mu} g^{\mathcal{E}}(\theta_j) + \frac{2}{\mu} N(\lambda_j^\dagger; \theta_{0:j}, \mathbf{w}_{0:j})\right] \quad (9)$$

*where* $p(I^{(\hat{T})} = j) \propto (1 - \mu\gamma)^{T \cdot (\hat{T}-1-j)}$ *for* $j \in \{0, \cdots, \hat{T} - 1\}$ *and* $\zeta_{\hat{T}} = \sum_{i=0}^{\hat{T}-1} (1 - \mu\gamma)^{T \cdot i}$.

Corollary 3.2.1 is proved in Appendix B.4 and gives a whole picture of the optimality gap under `AnCon`. We first remark the trade-off associated with $\hat{T}$ on the suboptimality $\mathbb{E}[l(\theta_{\hat{T}}) - l^*]$, which characterize the early-learning phenomenon observed in the biased gradient settings (e.g., self-training [31] and LFN [32]). Specifically, in (9), increasing $\hat{T}$ reduces the first term $(1-\mu\gamma)^{T \cdot (\hat{T}-1)}(l(\theta_0) - l^*)$ but increases the coefficient of the second term $\zeta_{\hat{T}}$. Therefore, a longer training with a large $\hat{T}$ may not enhance the self-training performance especially under a large second term due to inaccurate pseudo labels or the temporal ensemble.

Nevertheless, for each outer loop iteration, $g^{\mathcal{E}}(\theta_j)$ would be smaller under `AnCon` than its value under vanilla self-training as $\mathbb{E}[l(\theta_j) - l^*]$ has a tighter upper bound under `AnCon` due to Theorem 3.2. Therefore, with the guarantee $N(\lambda_j^\dagger; \theta_{0:j}, \mathbf{w}_{0:j}) \leq N(0)$, `AnCon` would achieve a tighter upper bound of (9) than the vanilla self-training method, enabling longer training with smaller value of the second term as observed in Figure 1b. Finally, we remark that this theoretical superiority of `AnCon` can be extended to the self-training methods with other weighting mechanisms in the asymptotic region when $\mathbb{E}_\xi[\langle \nabla \tilde{l}_\xi(\theta^*), \hat{g}_\xi(\theta_{0:m}, \mathbf{w}_{0:m}) \rangle] \geq 0$ (cf. Theorem 3.2).

## 4 Experiments

**Goal and baselines** Part of the experiments shows that `AnCon` surpasses the vanilla self-training method that uses $\hat{Y}(x; \theta)$ as a pseudo label, which serves as a strong baseline, under different types

Table 1: SFDA benchmark results. The numbers indicate the mean test accuracy across three repetitions. We present half of the domain pairs for Office-31 and OfficeHome in the main body and the rest of the pairs are presented in Appendix (cf. Tables 3, 4, and 6).

| Method | Office-31 | | | | OfficeHome | | | | | | | VisDa |
|---|---|---|---|---|---|---|---|---|---|---|---|---|
| | A2D | A2W | D2A | Avg (all) | Ar2Rw | Ar2Pr | Ar2Cl | Rw2Ar | Rw2Pr | Rw2Cl | Avg (all) | VisDa |
| Self-Training | 79.32 | 80.88 | 62.07 | 80.47 | 75.19 | 69.20 | 43.07 | 66.13 | 78.10 | 46.64 | 61.55 | 67.77 |
| + ELR ($\lambda = \lambda^*_{ELR}$) | 79.02 | **81.19** | 63.70 | 80.84 | 75.97 | 70.47 | 45.80 | 67.00 | 78.64 | 50.01 | 63.03 | **71.89** |
| + AnCon | **82.23** | 79.94 | **63.99** | **81.37** | **76.13** | **70.56** | **48.06** | **67.57** | **79.27** | **51.89** | **63.91** | 71.11 |
| GCE | **86.85** | 86.16 | 64.63 | 83.16 | 74.65 | 69.69 | 43.46 | 68.87 | 79.25 | 49.18 | 63.00 | 65.20 |
| + ELR ($\lambda = \lambda^*_{ELR}$) | **86.85** | 86.54 | 64.80 | 83.20 | 74.89 | 71.62 | 43.45 | 69.63 | 79.98 | 49.68 | 63.77 | 66.90 |
| + AnCon | 86.75 | **87.11** | **65.78** | **83.50** | **76.47** | **71.80** | **44.83** | **70.21** | **80.15** | **51.74** | **64.81** | **68.40** |
| NRC | 92.67 | 88.11 | 72.83 | 87.42 | 73.86 | 75.04 | 47.90 | 61.15 | 76.59 | 51.75 | 63.92 | 74.30 |
| + ELR ($\lambda = \lambda^*_{ELR}$) | 92.77 | 87.92 | 72.77 | 87.46 | 76.82 | 76.01 | 52.19 | 66.01 | **80.40** | 55.69 | 66.89 | 82.80 |
| + AnCon | **94.28** | **91.45** | **74.49** | **89.07** | **79.64** | **77.11** | **53.56** | **69.47** | 80.04 | **57.30** | **67.96** | **83.70** |

of distribution shifts (Sections 4.1-4.2). We also show that `AnCon` achieves a better performance than ELR with $\lambda^*_{ELR} \in \{1, 3, 7, 12, 25\}$, which is the coefficient multiplied to the auxiliary loss, to show effectiveness of uncertainty aware temporal ensemble. Finally, we assess the integration of `AnCon` with generalized cross-entropy (GCE) [33, 31] and neighborhood reciprocity clustering (NRC) [17], which are frequently cited as state-of-the-art methods [20, 34, 31]. We note that `AnCon` can be seamlessly applied to GCE and NRC by replacing one-hot pseudo labels by `AnCon`'s regularized pseudo labels $\hat{Y}(X; \theta_{0:m}, \mathbf{w}_{0:m})$. For descriptions of the training configurations which we adopt from literature, see Appendix D.

**Evaluation** In our self-training settings (SFDA and TTA), we should determine the best checkpoint without labeled samples, i.e. to select a model $\theta_0, \theta_1, \cdots, \theta_I$ which is used for evaluation on test. We use information maximization [35], $IM(\theta) = H_E(\mathbb{E}_X[f(X; \theta)]) - \mathbb{E}_X[H_E(f(X; \theta))]$ when $H_E$ is the entropy, which is proven to be effective in unsupervised domain adaptation (UDA) model selection literature [36, 37]. Specifically, we evaluate $IM(\theta_m)$ on the hold-out unlabeled samples at the end of each epoch $m$ for $I$ number of training epochs and select the checkpoint with $\theta^* \in \arg\max_{m \in [I]} IM(\theta_m)$ as the best one. This $\theta^*$ is used for final evaluation on the test data.

**Implementation details of** `AnCon` For each (outer) iteration $m$, we set $w_j(x) = \mathbf{1}(c(x; \theta_j) > \delta_j^{(\beta)})$ for $j \in \{0, \cdots, m\}$, instead of $w_j(x) = \frac{\mathbf{1}(c(x; \theta_j) > \delta_j^{(\beta)})}{\sum_{i=0}^{m} \mathbf{1}(c(x; \theta_i) > \delta_i^{(\beta)})}$. Note that the former with $\lambda$ can be thought of as the latter with instance-dependent coefficient $\lambda \cdot Q(x; \mathbf{c}_{0:m})$, which assigns more weight on the generalized temporal ensemble for $x$ with large $Q(x; \mathbf{c}_{0:m})$. In addition, we consider the online update scenario of pseudo labels, i.e., $T = 1$. Finally, given the challenging nature of hyperparameter selection in self-training under distribution shifts, we perform all experiments by using a single configuration of hyperparameters of `AnCon` ($\lambda = 0.3$ and $\beta = 0.9$). These hyperparameters result in the maximum value of $\max_{m \in [\hat{T}]} IM(\theta_m)$ on the hold-out unlabeled samples on Office-31.

## 4.1 Self-training under domain shifts

We first consider SFDA that adapts a model trained in one domain by performing self-training in the distribution shifted domain. For the network architecture, we use the modified ResNet used in [16], which includes batch normalization [38] after the bottleneck layer and weight normalization [39] in the last linear layer, which is used in [17, 8] for stabilizing the learning process in SFDA.

**Datasets** We evaluate `AnCon` on the following datasets: **Office-31** [40] with 4,000 images of 31 categories from three domains (amazon, dslr, webcam); **OfficeHome** [41] with 15,000 images of 65 categories from four domains (art, clipart, product, real-world); **VisDa** [42] with 280,000 images of 31 categories from two domains (synthetic, real).

**Results** `AnCon` consistently improves self-training in diverse domain pairs across the three datasets (Table 1). Specifically, it reduces the average self-training test error by 5% in Office-31, 6% in OfficeHome, and 13% in VisDa. In addition, compared to ELR with its dataset-dependent optimal hyperparameter value, `AnCon` shows comparable performance to ELR (2%, 3%, and -3% performance differences in Office-31, OfficeHome, and VisDa, respectively). Despite its slightly inferior performance in VisDa, we note that `AnCon` achieves significantly better accuracy for low-performing classes: for "Skateboard" and "Truck" classes, `AnCon` achieves 50% and 16% of accuracies, while ELR achieves 39% and 0.43% in these classes. Thus, `AnCon` can be as effective as ELR for improving the self-training performances under distribution shifts without needing to adjust hyperparameter values for each dataset unlike ELR.

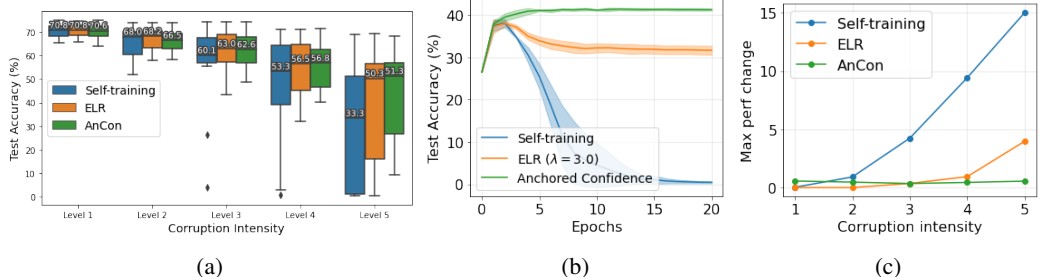

(a)                                          (b)                                          (c)

Figure 1: **Section 4.2:** (a) Test accuracy for each intensity level in ImageNet-C. (b) Performance degeneration in the defocus blur corruption with intensity 4. **Section 4.3.1:** (c) Maximum performance changes under different model selection methods. We present performances for individual corruptions in Appendix. For all boxplots used in the paper, the box represents interquantile range with whiskers as $\pm$ 1.5 interquantile range and the horizontal line inside the box represents the median.

Notably, `AnCon` significantly improves performances of both GCE and NRC via fundamentally different mechanisms for handling noisy pseudo labels (e.g., reducing test accuracy of 3% and 8% on average in OfficeHome, respectively). Specifically, NRC filters incorrect predictions based on local consistency, while `AnCon` uses temporal consistency. Combining NRC and `AnCon` leverages pseudo labels that are both locally and temporally consistent, resulting in significant performance improvements over NRC or Self-Training + `AnCon` (cf. Table 1). In addition, GCE reduces the impact of wrong pseudo labels rather than finding them. Applying GCE to `AnCon` minimizes the effects of potentially wrong but temporally consistent pseudo labels, which can be implied by the performance of GCE + `AnCon` compared to GCE or Self-Training + `AnCon` (cf. Table 1). Thus, the impressive performance gains from `AnCon`, which would be orthogonal to the gains from state-of-the-art methods in SFDA, show its significant practical implications.

## 4.2 Self-training under synthetic corruption operations

While we have considered the domain shift, e.g., adaptation of a model trained on synthetic images to real images, in Section 4.1, this section examines the self-training's ability to adapt to distribution shifts by synthetic image corruptions. This setting has been used to measure the robustness of neural networks with respect to a general out-of-distribution setting. To this end, we consider **ImageNet-C** [6], which consists of 50,000 images drawn from a validation set of ImageNet [43] where each image is corrupted by 15 types of synthetic corruptions related to noise, blur, weather and digital.

**Result** Consistent with the findings under the domain shift, `AnCon` outperforms the average performances of self-training and ELR under varying levels of corruption intensities (cf. Figure 1a and Tables 7-11 in Appendix), improving the self-training method's accuracy by 16% on average. Further, the gains from `AnCon` is significant when the distribution shifts are intense (e.g., improving accuracies by 20% and 52% on average in intensities of 4 and 5) where the initial model trained on the source domain significantly deteriorates. Specifically, for Shot, Impulse, and Gaussian corruptions with the most extreme shift intensity of 5, where the initial model achieves accuracies of (3.04%, 1.76%, 2.12%), `AnCon` achieves (22.56%, 26.56%, 25.85%) (cf. Table 11). This striking improvement compared to vanilla self-training with performances (0.26%, 1.72%, 1.04%) and ELR with performances (8.00%, 14.12%, 16.00%), underscores the importance of the `AnCon`'s uncertainty-aware temporal consistency scheme, as shown in Corollary 3.2.1. We note that this impressive result is also explained by `AnCon`'s ability to prevent the gradual performance degradation during the course of training with the extremely noisy pseudo labels (cf. Figure 1b). Combined with previous results in domain shift scenarios, we expect that `AnCon` would work effectively in various out-of-distribution settings.

## 4.3 Versatility of AnCon

In previous sections, we have shown the universality of `AnCon` by evaluating it on diverse distribution shift scenarios. In this section, we show versatility of `AnCon` by analyzing its attractive properties in robustness and uncertainty representation.

### 4.3.1 Robustness to model selection

There is no universally agreed model selection criterion, such as cross-validation in the i.i.d. setting, in self-training under distribution shifts. This is partly due to the variety of distribution shift scenarios, where an effective criterion in one may be ineffective or inapplicable in another; for instance, a

principled criteria called importance-weighted cross validation [44] in UDA cannot be applied to SFDA. In this regard, it would be an important characteristic of a self-training method under distribution shift to be robust with respect to different choices of model selection criteria. Therefore, we evaluate robustness with respect to the following different model selection criteria: InfoMax [35], Corr-C [36], and Ent [45] (see Appendix D.3 for the description).

Figure 1c shows that `AnCon`'s maximum performance change due to different model selection methods is much lower than that of other methods, especially under severe distribution shifts. This valuable advantage can be contributed to the property of `AnCon` that can prevent performance degeneration (cf. Figure 1b). Given that, in practice, we barely know when the model collapse happens and which model selection criteria are the best, the results highlight a significant practical value of `AnCon`.

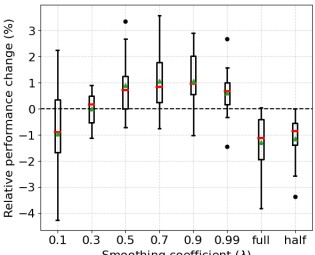

(a)

### 4.3.2 Robustness to the choice of hyperparameters

Throughout this paper, we have shown that our single configuration of parameters ($\lambda = 0.3, \beta = 0.9$) work well across a wide range of benchmark problems. In this section, we aim to show our findings can be preserved when the hyperparameter values deviate from the default setting by performing a sensitivity analysis for values $\lambda \in \{0.1, 0.3, 0.5, 0.7, 0.9\}$ and $\beta \in \{0.1, 0.3, 0.5, 0.7, 0.9\}$. We also test two frequently used annealing schedules that $\lambda_m = m/I$ and $\lambda_m = \min(1, 2m/I)$, called full and half, respectively. Figure 2 shows that `AnCon` is stable even under extreme values of hyperparameters. Specifically, for both hyperparameters, the maximum average performance change is less than 1%, and $\beta$ barely impacts the performance of `AnCon`. Indeed, our analysis suggests to increase $\lambda$ from our default setting; that is, to put a higher weight on the general temporal ensemble's prediction. Here, we note that our suboptimal choices of hyperparameters are due to our rigorous and practical hyperparameter choice. Given the challenging nature of hyperparameter optimization under distribution shifts, the stable performances of `AnCon` under arbitrary choices of hyperparameters would enable `AnCon` to be seamlessly applied to diverse practical settings.

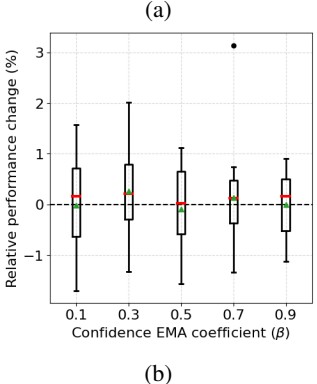

(b)

Figure 2: Sensitivity analysis with respect to $\lambda$ and $\beta$ on four domain pairs (Ar-Pr, Pr-Cl, Rw-Cl, Rw-Pr) in OfficeHome. Here, green triangles are means.

### 4.3.3 Improved calibration performance

We have shown that all self-training methods significantly improve the performance of the baseline method after the adaptation period. However, it is widely known that these noticeable improvements come with the price of sacrificing an uncertainty representation ability which is critical in real-world decision-making scenarios [46, 47]. Specifically, the calibration performance, which is the gap between the prediction confidence and accuracy, usually monotonically increases as self-training keeps reducing the uncertainty for all predictions during the course of training. In this regard, we analyze the calibration performance with respect to the expected calibration error (ECE; see Appendix for definition). Here, a lower ECE means a lower gap between confidence and accuracy.

As shown in Figure 3a and Table 5 in Appendix, `AnCon` gives much lower ECE compared to other methods. Considering ELR and GCE both have regularization effects, we conjecture that this phenomenon is due to selective regularization in `AnCon` that increases prediction confidences of samples only if the past confident predictions are consistent with the current prediction. Especially, in

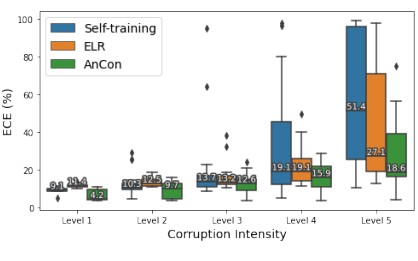

(a)

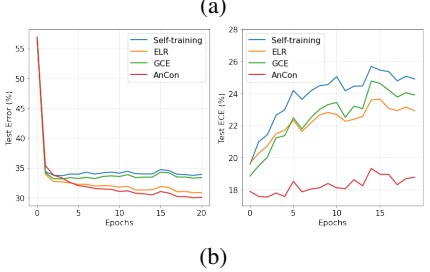

(b)

Figure 3: (a) ECEs under five levels of intensities in ImageNet-C; (b) Accuracy and ECE changes during the course of training in VisDa.

Figure 3b which confirms the accuracy-calibration dilemma in VisDa, `AnCon` is shown to limit the ECE increases during training compared to all other methods. That is, `AnCon` helps to significantly reduce the price of the calibration performance we need to pay for improving accuracy, which are both important measures in practice.

### 4.4 Algorithmic design choices

Recall that we define $\bar{f}(x; \theta_{0:m}, \mathbf{w}_{0:m}) = \sum_{i=0}^{m} w_i(x) \cdot p(y|x, \theta_i)$ with our simple design choices: the relative thresholding for weighting scheme $w_i(x) \propto \mathbf{1}(c(x; \theta_i) > \delta_i^{(\beta)})$ and hard prediction for $p(y|x, \theta_i)$. In Appendix C.3, we found that our simple design choices are more appropriate for the distribution shift settings than several more sophisticated alternatives, which can be summarized as follows.

- More sophisticated weighting schemes (e.g., Entropy ($w_i(x) \propto \exp\{-H_E[f(x; \theta_i)]\}$)) reduce the self-training performance, despite being a more accurate measure of prediction uncertainty. We conjecture that the poor calibration performance of the neural network in self-training under distribution shifts prevents the sophisticated weighting schemes from accurately reflecting the goodness of the prediction.
- Various soft prediction schemes, which can give more information about the non-leading entry values, leads to performance reductions. We conjecture that the continuously increasing confidence in the later stage of self-training would make soft prediction ignore early-stage predictions which may be valuable to memorize.

## 5 Related work

**Filtering incorrect pseudo labels** Popular confidence-based thresholding methods [13, 14, 9] fall short under distribution shifts since even high confident predictions can be highly incorrect. Therefore, recent advances in SFDA and TTA utilize higher order information to filter incorrect pseudo labels. For instance, based on the intuition that true labels of adjacent samples would be same, centroids for each predicted class can be maintained in the feature space and then the pseudo label for each input is corrected by the adjacent centroid [16]. The idea of using per-class centroids has been extended to incorporate more general clustering structures [18, 17]. However, the neighborhood structure-based methods are computationally demanding due to storage of memory banks in the feature space and nearest neighbors search. Such computational complexity persists in other approaches, which are based on the consistency of multiple predictions from different augmentations [20] and models trained with different loss functions [19]. Compared to these solutions, `AnCon` can efficiently estimate correct labels with only limited extra memory overhead of storing past predictions.

**Learning from noisy labels** Treating pseudo labels as inherently noisy, techniques from the LFN literature have been integrated to self-training. For instance, the LFN literature has proposed robust loss functions that reduce impacts of random noisy labels [48, 33], and a recent large-scale experimental study shows the applicability of the generalized cross-entropy in the SFDA setting [31]. The effectiveness of ELR on SFDA [8] bears a similar idea because ELR was developed to regularize the neural networks' tendencies to memorize incorrect labels [21]. Despite their effectiveness, by nature, these approaches do not consider important characteristics of the unbounded and instance-dependent noise rates inherent in self-training under distribution shifts, which results in significant suboptimality in both theory and practice. However, by considering the unique characteristics of self-training under distribution shift, `AnCon` relaxes the conditions required to achieve optimality as well as boosts the self-training performance in diverse scenarios.

## 6 Conclusion

This paper introduces `AnCon`, which effectively improves self-training performances under diverse distribution shift scenarios by promoting selective temporal consistency based on confident predictions. As a result, `AnCon` effectively mitigates the detrimental effects of noisy pseudo labels without much computational overhead, unlike the previous methods. We show that `AnCon` not only advances our theoretical understanding of a generalized notion of temporal consistency in self-training but also can be a practical asset as a simple and effective self-training method with attractive properties. In Appendix C.4, we present limitations and future directions, such as adaptive determination of $\lambda$, combining local and temporal consistency, and extending the selective temporal consistency in the sequential decision making problems.

## Acknowledgments and Disclosure of Funding

We would like to thank Jihyeon Hyeong, Yuchen Lou, Jiezhong Wu, and anonymous reviewers for their valuable discussions and constructive suggestions during the preparation of this manuscript. We declare that there was no funding received for this work.

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

# A List of acronyms

| | |
|---|---|
| TTA | Test time adaptation |
| SFDA | Source-free domain adaptation |
| i.i.d. | independent and identically distributed |
| ELR | Early learning regularization |
| LFN | Learning from noisy labels |
| KD | Knowledge distillation |
| EMA | Exponential moving average |
| GCE | Generalized cross-entropy |
| NRC | Neighborhood reciprocity clustering |
| PL | Polyak-Lojasiewicz |

# B Proof of claims

## B.1 Assumptions

Let us recall that $l(\theta) = \mathbb{E}_{XY}[H(f(X;\theta), p_{Y|X})]$ and $l_\xi(\theta) = \frac{1}{b}\sum_{X_i \in \xi} H(f(X_i;\theta), p_{Y|X_i})$ where $\xi$ contains $b$ number of random samples from $p_X$. We also denote $\theta^* \in \arg\min_{\theta \in \mathbb{R}^p} l(\theta)$. In this work, we assume the following three regularity conditions on $l(\theta)$ which is differentiable with respect to $\theta$, which are mild but essential for most theoretical studies with convergence analyses.

**Assumption B.1** ($L$-smoothness). $l(\cdot)$ *is L-smooth for some constant* $L > 0$*; that is,*

$$l(\theta') \leq l(\theta) + \langle \nabla l(\theta), \theta' - \theta \rangle + \frac{L}{2} \parallel \theta' - \theta \parallel^2, \quad \forall \theta \in \mathbb{R}^p \text{ and } \theta' \in \mathbb{R}^p. \tag{10}$$

**Assumption B.2** ($\mathcal{L}$-expected smoothness [30]). $l(\cdot)$ *is* $\mathcal{L}$*-smooth in expectation with respect to* $\xi \sim \mathcal{D}$*; that is,*

$$\mathbb{E}_{\xi \sim \mathcal{D}}[\parallel \nabla l_\xi(\theta) - \nabla l_\xi(\theta^*) \parallel^2] \leq 2\mathcal{L}(l(\theta) - l(\theta^*))), \quad \forall \theta \in \mathbb{R}^p. \tag{11}$$

**Assumption B.3** ($\mu$-Polyak-Lojasiewicz (PL) condition [49]). $l(\cdot)$ *satisfies the* $\mu$*-PL condition for some constant* $\mu > 0$*; that is,*

$$\parallel \nabla l(\theta) \parallel^2 \geq 2\mu(l(\theta) - l(\theta^*)), \quad \forall \theta \in \mathbb{R}^p. \tag{12}$$

We remark that Assumptions B.1 and B.2 can trivially hold under bounded parameter values that can be guaranteed by optimizing neural networks with finite iterations under a gradient or weight clipping. Assumption B.3 holds for infinite-width neural networks, i.e., the neural tangent kernel (NTK) regime [50]. Given that the gradient descent training dynamics of neural networks can be well approximated by NTK [51], the PL condition can be generally regarded as a mild assumption.

## B.2 Proof of Theorem 3.1

*Proof.* For $x$ such that $Q(x; \mathbf{c}_{0:m}) > 0$, let us consider subsequence of index $l$ such that $x \in A_l(c_l)$; that is, $(j_1, \cdots, j_{Q(x;\mathbf{c}_{0:m})})$ where $j_l = \min\{l \geq j_{l-1} | x \in A_l(c_l)\}$ and $j_0 = 0$. Let $S_{Q(x;\mathbf{c}_{0:m})} = \sum_{i=1}^{Q(x;\mathbf{c}_{0:m})} \mathbf{1}(Y(x) = \hat{Y}(x; \theta_{j_i}))$. Then, we have the following inequality

$$p\left(\arg\max_k \bar{f}_k(x; \theta_{0:m}, \mathbf{w}_{0:m}) \neq Y(x)\right) \leq p\left(S_{Q(x;\mathbf{c}_{0:m})} \leq \frac{Q(x; \mathbf{c}_{0:m})}{2}\right)$$

$$\leq \exp\left(-\frac{Q(x; \mathbf{c}_{0:m})}{2} \cdot (2\bar{p}(x; \mathbf{c}_{0:m}) - 1 - \log(2\bar{p}(x; \mathbf{c}_{0:m})))\right) \tag{13}$$

where the first inequality holds because $\arg\max_k \bar{f}_k(x; \theta_{0:m}, \mathbf{w}_{0:m}) = Y(x)$ if $S_{Q(x;\mathbf{c}_{0:m})} > Q(x; \mathbf{c}_{0:m})/2$ and the second inequality holds due to Lemma B.4 given later in Appendix B.5 with $p_i = p(Y(x) = \hat{Y}(x; \theta_i))$, $o = Q(x; \mathbf{c}_{0:m})$, and $q = 1/2$. $\qquad\square$

## B.3 Proof of Theorem 3.2

*Proof.* By Lemma B.5, any realization of $\theta_m$ satisfies

$$\mathbb{E}[l(\theta_{m,t}) - l(\theta^*)|\theta_m] \leq (1 - \gamma\mu)^t(l(\theta_m) - l(\theta^*)) + \frac{8C^2}{\mu}g^{\mathcal{E}}(\theta_m)$$

$$+ \frac{2}{\mu}\left(\lambda^2 \parallel \hat{g}(\theta_{0:m}, \mathbf{w}_{0:m}) \parallel^2 + \frac{L\gamma}{2}\mathbb{E}_\xi\left[\parallel \nabla\tilde{l}_\xi(\theta^*) - \lambda\hat{g}_\xi(\theta_{0:m}, \mathbf{w}_{0:m}) \parallel^2\right]\right). \quad (14)$$

Then, from the definition $N(\lambda; \theta_{0:m}, \mathbf{w}_{0:m}) = \lambda^2 \parallel \hat{g}(\theta_{0:m}, \mathbf{w}_{0:m}) \parallel^2 + \frac{L\gamma}{2}\mathbb{E}_\xi\left[\parallel \nabla\tilde{l}_\xi(\theta^*) - \lambda\hat{g}_\xi(\theta_{0:m}, \mathbf{w}_{0:m}) \parallel^2\right]$, we get (7). To show $N(\lambda_m^\dagger; \theta_{0:m}, \mathbf{w}_{0:m}) \leq N(0)$, we apply Lemma B.6 as

$$\frac{N(\lambda_m^\dagger; \theta_{0:m}, \mathbf{w}_{0:m})}{N(0)} = 1 - \frac{\left(\mathbb{E}_\xi[\langle\nabla\tilde{l}_\xi(\theta^*), \hat{g}_\xi(\theta_{0:m}, \mathbf{w}_{0:m})\rangle]\right)^2}{\left(\mathbb{E}_\xi \parallel \hat{g}_\xi(\theta_{0:m}, \mathbf{w}_{0:m}) \parallel^2 + \frac{2}{L\gamma} \parallel \hat{g}(\theta_{0:m}, \mathbf{w}_{0:m}) \parallel^2\right)\left(\mathbb{E}_\xi \parallel \nabla\tilde{l}_\xi(\theta^*) \parallel^2\right)}.$$
$$(15)$$

To show (8), let us assume $\mathbb{E}_\xi[\langle\nabla\tilde{l}_\xi(\theta^*), \hat{g}_\xi(\theta_{0:m}, \mathbf{w}_{0:m})\rangle] \geq 0$. Then, we get the following inequality:

$$\frac{\mathbb{E}_\xi[\langle\nabla\tilde{l}_\xi(\theta^*), \hat{g}_\xi(\theta_{0:m}, \mathbf{w}_{0:m})\rangle]}{(\mathbb{E}_\xi \parallel \nabla\tilde{l}_\xi(\theta^*) \parallel^2)^{1/2} \cdot (\mathbb{E}_\xi \parallel \hat{g}_\xi(\theta_{0:m}, \mathbf{w}_{0:m}) \parallel^2)^{1/2}} \quad (16)$$

$$= \frac{\mathbb{E}_\xi \parallel \nabla\tilde{l}_\xi(\theta^*) \parallel^2 + \mathbb{E}_\xi \parallel \hat{g}_\xi(\theta_{0:m}, \mathbf{w}_{0:m}) \parallel^2 - \mathbb{E}_\xi \parallel \nabla\tilde{l}_\xi(\theta^*) - \hat{g}_\xi(\theta_{0:m}, \mathbf{w}_{0:m}) \parallel^2}{2(\mathbb{E}_\xi \parallel \nabla\tilde{l}_\xi(\theta^*) \parallel^2)^{1/2} \cdot (\mathbb{E}_\xi \parallel \hat{g}_\xi(\theta_{0:m}, \mathbf{w}_{0:m}) \parallel^2)^{1/2}} \quad (17)$$

$$\geq 1 - \min\left(1, \frac{\mathbb{E}_\xi \parallel \nabla\tilde{l}_\xi(\theta^*) - \hat{g}_\xi(\theta_{0:m}, \mathbf{w}_{0:m}) \parallel^2}{2(\mathbb{E}_\xi \parallel \nabla\tilde{l}_\xi(\theta^*) \parallel^2)^{1/2} \cdot (\mathbb{E}_\xi \parallel \hat{g}_\xi(\theta_{0:m}, \mathbf{w}_{0:m}) \parallel^2)^{1/2}}\right) \quad (18)$$

$$\geq 1 - \min\left(1, \frac{C^2\mathbb{E}_X D_{KL}(f(X; \theta^*) \parallel \bar{f}(X; \theta_{0:m}, \mathbf{w}_{0:m}))}{(\mathbb{E}_\xi \parallel \nabla\tilde{l}_\xi(\theta^*) \parallel^2)^{1/2} \cdot (\mathbb{E}_\xi \parallel \hat{g}_\xi(\theta_{0:m}, \mathbf{w}_{0:m}) \parallel^2)^{1/2}}\right) \quad (19)$$

where (18) holds due to $a^2 + b^2 \geq 2ab$; (19) holds due to the assumption $\parallel x \parallel \leq C$, $\parallel \cdot \parallel_2 \leq \parallel \cdot \parallel_1$, and the Pinsker's inequality applied as

$$\mathbb{E}_\xi \parallel \nabla\tilde{l}_\xi(\theta^*) - \hat{g}_\xi(\theta_{0:m}, \mathbf{w}_{0:m}) \parallel^2 \leq C^2\mathbb{E}_\xi \frac{1}{b}\sum_{X_i \in \xi} \parallel f(X_i; \theta^*) - \bar{f}(X_i; \theta_{0:m}, \mathbf{w}_{0:m}) \parallel^2$$

$$\leq C^2\mathbb{E}_\xi \frac{1}{b}\sum_{X_i \in \xi} \parallel f(X_i; \theta^*) - \bar{f}(X_i; \theta_{0:m}, \mathbf{w}_{0:m}) \parallel_1^2$$

$$\leq 2C^2\mathbb{E}_X D_{KL}(f(X; \theta^*), \bar{f}(X; \theta_{0:m}, \mathbf{w}_{0:m})). \quad (20)$$

Thus, we get our desired result by combining the following inequality with (15):

$$\frac{N(\lambda_m^\dagger; \theta_{0:m}, \mathbf{w}_{0:m})}{N(0)} = 1 - \frac{\left(\mathbb{E}_\xi[\langle\nabla\tilde{l}_\xi(\theta^*), \hat{g}_\xi(\theta_{0:m}, \mathbf{w}_{0:m})\rangle]\right)^2}{\left(\mathbb{E}_\xi \parallel \hat{g}_\xi(\theta_{0:m}, \mathbf{w}_{0:m}) \parallel^2 + \frac{2}{L\gamma} \parallel \hat{g}(\theta_{0:m}, \mathbf{w}_{0:m}) \parallel^2\right)\left(\mathbb{E}_\xi \parallel \nabla\tilde{l}_\xi(\theta^*) \parallel^2\right)}$$
$$(21)$$

$$= 1 - \left(\frac{\left(\mathbb{E}_\xi[\langle\nabla\tilde{l}_\xi(\theta^*), \hat{g}_\xi(\theta_{0:m}, \mathbf{w}_{0:m})\rangle]\right)^2}{(\mathbb{E}_\xi \parallel \nabla\tilde{l}_\xi(\theta^*) \parallel^2)^{1/2} \cdot (\mathbb{E}_\xi \parallel \hat{g}_\xi(\theta_{0:m}, \mathbf{w}_{0:m}) \parallel^2)^{1/2}}\right)^2 \frac{1}{1 + \frac{2}{L\gamma}\frac{\|\hat{g}(\theta_{0:m}, \mathbf{w}_{0:m})\|^2}{\mathbb{E}_\xi\|\hat{g}_\xi(\theta_{0:m}, \mathbf{w}_{0:m})\|^2}} \quad (22)$$

$$\leq 1 - \left(1 - \min\left(1, \frac{C^2 \mathbb{E}_X D_{KL}(f(X;\theta^*) \,\|\, \bar{f}(X;\theta_{0:m}, \mathbf{w}_{0:m}))}{(\mathbb{E}_\xi \| \nabla \tilde{l}_\xi(\theta^*) \|^2)^{1/2} \cdot (\mathbb{E}_\xi \| \hat{g}_\xi(\theta_{0:m}, \mathbf{w}_{0:m}) \|^2)^{1/2}}\right)\right)^2 \frac{1}{1 + \frac{2}{L\gamma} \frac{\|\hat{g}(\theta_{0:m}, \mathbf{w}_{0:m})\|^2}{\mathbb{E}_\xi \|\hat{g}_\xi(\theta_{0:m}, \mathbf{w}_{0:m})\|^2}} \tag{23}$$

$$\leq 2 \min\left(1, \frac{C^2 \mathbb{E}_X D_{KL}(f(X;\theta^*) \,\|\, \bar{f}(X;\theta_{0:m}, \mathbf{w}_{0:m}))}{(\mathbb{E}_\xi \| \nabla \tilde{l}_\xi(\theta^*) \|^2)^{1/2} \cdot (\mathbb{E}_\xi \| \hat{g}_\xi(\theta_{0:m}, \mathbf{w}_{0:m}) \|^2)^{1/2}}\right) + \frac{2}{L\gamma} \frac{\| \hat{g}(\theta_{0:m}, \mathbf{w}_{0:m}) \|^2}{\mathbb{E}_\xi \| \hat{g}_\xi(\theta_{0:m}, \mathbf{w}_{0:m}) \|^2} \tag{24}$$

$$\leq 2 \min\left(1, \frac{C^2 \mathbb{E}_X D_{KL}(f(X;\theta^*) \,\|\, \bar{f}(X;\theta_{0:m}, \mathbf{w}_{0:m}))}{(\mathbb{E}_\xi \| \nabla \tilde{l}_\xi(\theta^*) \|^2)^{1/2} \cdot (\mathbb{E}_\xi \| \hat{g}_\xi(\theta_{0:m}, \mathbf{w}_{0:m}) \|^2)^{1/2}}\right) \tag{25}$$

$$+ \frac{2C^2}{L\gamma} \frac{\| \mathbb{E}_X[\bar{f}(X;\theta_{0:m}, \mathbf{w}_{0:m})] - \mathbb{E}_X[\hat{Y}(X;\theta_m)] \|^2}{\mathbb{E}_\xi \| \hat{g}_\xi(\theta_{0:m}, \mathbf{w}_{0:m}) \|^2} \tag{26}$$

where (24) holds due to $1 - \frac{(1-u)^2}{1+v} \leq 2u + v$ [30]. $\qquad\square$

## B.4 Proof of Corollary 3.2.1

*Proof.* For any realization of $\theta_m$ for $m \in \{0, 1, \cdots, \hat{T} - 1\}$, applying Theorem 3.2 with $\lambda = \lambda_m^\dagger$ (cf. Lemma B.6) gives

$$\mathbb{E}[l(\theta_{m+1}) - l^* | \theta_m] = \mathbb{E}[l(\theta_{m,T}) - l^* | \theta_m]$$

$$\leq (1 - \mu\gamma)^T (l(\theta_m) - l^*) + \frac{8C^2}{\mu} g^{\mathcal{E}}(\theta_m) + \frac{2}{\mu} N(\lambda_m^\dagger; \theta_{0:m}, \mathbf{w}_{0:m}). \tag{27}$$

Thus, by recursively applying (27) to $\theta_0$, we get our desired result as

$$\mathbb{E}[l(\theta_{\hat{T}}) - l^*] \leq (1 - \mu\gamma)^{T \cdot (\hat{T} - 1)}(l(\theta_0) - l^*) + \sum_{i=0}^{\hat{T}-1} (1 - \mu\gamma)^{T \cdot (\hat{T} - 1 - i)} \left(\frac{8C^2}{\mu} g^{\mathcal{E}}(\theta_i) + \frac{2}{\mu} N(\lambda_i^\dagger; \theta_{0:i}, \mathbf{w}_{0:i})\right). \tag{28}$$

$\qquad\square$

## B.5 Supporting lemmas

**Lemma B.4.** *Let $A_i \sim Bern(p_i)$ with $p_i \in [0, 1]$ and $S_o = \sum_{i \in [o]} A_i$, where $A_i$ and $A_j$ are independent for $i \neq j$. Then for any $q < \bar{p} := \frac{1}{o} \sum_{i \in [o]} p_i$, we have the tail probability bound*

$$p(S_o \leq q \cdot o) \leq \exp\left(o\left(q - \bar{p} - q \log \frac{q}{\bar{p}}\right)\right). \tag{29}$$

*Proof.* For any $t$, the moment generating function of $S_m$ is given by

$$M(t) = \mathbb{E}_{S_o}[\exp(tS_o)] = \prod_{i \in [o]} \mathbb{E}_{A_i}[\exp(tA_i)] = \exp\left(\sum_{i \in [o]} \log(\mathbb{E}_{A_i}[\exp(tA_i)])\right)$$

$$= \exp\left(\sum_{i \in [o]} \log(p_i(\exp(t) - 1) + 1)\right) \leq \exp\left(\sum_{i \in [o]} p_i(\exp(t) - 1)\right) = \exp(o\bar{p}(\exp(t) - 1)) \tag{30}$$

where the first equality comes from the independence and the inequality is from $\log x \leq x - 1$.

Then, for $t < 0$ and $q < \bar{p}$, the Chernoff bound [52] gives

$$p(S_o \leq q \cdot o) \leq M(t)\exp(-t \cdot q \cdot o) = \exp(o\bar{p}(\exp(t) - 1) - t \cdot q \cdot o). \tag{31}$$

Finally, setting $t = \log(q/\bar{p}) < 0$ gives our desired result. $\qquad\square$

**Lemma B.5** (Modification from [30]). *Let us assume that $l(\theta)$ satisfies the L-smoothness, the $\mathcal{L}$-expected smoothness, and the $\mu$-PL condition. Also, we assume a linear model $f_k(x;\theta) = \{\exp(\theta^T x)\}_k / \sum_{i \in [K]} \{\exp(\theta^T x)\}_i$ with $\theta \in \mathbb{R}^{d \times K}$. Then, for the stochastic gradient descent $\theta_{m,t+1} = \theta_{m,t} - \gamma g_\xi^{(m,t)}$ with $g_\xi^{(m,t)} = \nabla \hat{\mathbb{E}}_{X_i \in \xi}[H(f(X_i;\theta_{m,t}), \tilde{Y}(X_i;\theta_{0:m}, \mathbf{w}_{0:m}))]$, $\theta_{m,0} := \theta_m$, and a constant learning rate $\gamma \leq \frac{\mu}{4\mathcal{L} \cdot L}$, any realization of $\theta_m$ satisfies*

$$\mathbb{E}[l(\theta_{m,t}) - l(\theta^*)|\theta_m] \leq (1 - \gamma\mu)^t (l(\theta_m) - l(\theta^*)) + \frac{8C^2}{\mu} g^{\mathcal{E}}(\theta_m)$$

$$+ \frac{1}{\mu} \left( 2\lambda^2 \parallel \hat{g}(\theta_{0:m}, \mathbf{w}_{0:m}) \parallel^2 + L\gamma \mathbb{E}_\xi \left[ \parallel \nabla \tilde{l}_\xi(\theta^*) - \lambda \hat{g}_\xi(\theta_{0:m}, \mathbf{w}_{0:m}) \parallel^2 \right] \right). \quad (32)$$

*Proof.* For any realization of $\theta_{m,t}$, we have the following bound:

$$\mathbb{E}_\xi[l(\theta_{m,t+1}) - l(\theta^*)|\theta_{m,t}] \leq (l(\theta_{m,t}) - l(\theta^*)) - \gamma\langle\nabla l(\theta_{m,t}), g^{(m,t)}\rangle + \frac{L\gamma^2}{2}\mathbb{E}_\xi\left[\parallel g_\xi^{(m,t)} \parallel^2\right]$$

$$\tag{33}$$

$$= (l(\theta_{m,t}) - l(\theta^*)) - \gamma\langle\nabla l(\theta_{m,t}), \nabla l(\theta_{m,t}) - b^{(m,t)}\rangle + \frac{L\gamma^2}{2}\mathbb{E}_\xi\left[\parallel \nabla\tilde{l}_\xi(\theta_{m,t}) - \lambda\hat{g}_\xi(\theta_{0:m}, \mathbf{w}_{0:m}) \parallel^2\right]$$

$$\tag{34}$$

$$= (l(\theta_{m,t}) - l(\theta^*)) - \gamma\parallel\nabla l(\theta_{m,t})\parallel^2 + \gamma\langle\nabla l(\theta_{m,t}), b^{(m,t)}\rangle$$

$$+ \frac{L\gamma^2}{2}\mathbb{E}_\xi\left[\parallel \nabla\tilde{l}_\xi(\theta_{m,t}) - \lambda\hat{g}_\xi(\theta_{0:m}, \mathbf{w}_{0:m}) \parallel^2\right]$$

$$\tag{35}$$

$$\leq (1 - \gamma\mu)(l(\theta_{m,t}) - l(\theta^*)) - \frac{\gamma\mu}{2}(l(\theta_{m,t}) - l(\theta^*)) + \gamma\parallel b^{(m,t)}\parallel^2$$

$$+ \frac{L\gamma^2}{2}\mathbb{E}_\xi\left[\parallel \nabla\tilde{l}_\xi(\theta_{m,t}) - \lambda\hat{g}_\xi(\theta_{0:m}, \mathbf{w}_{0:m}) \parallel^2\right]$$

$$\tag{36}$$

$$\leq (1 - \gamma\mu)(l(\theta_{m,t}) - l(\theta^*)) - \frac{\gamma\mu}{2}(l(\theta_{m,t}) - l(\theta^*)) + \gamma\parallel b^{(m,t)}\parallel^2$$

$$+ L\gamma^2\mathbb{E}_\xi\left[\parallel \nabla\tilde{l}_\xi(\theta_{m,t}) - \nabla\tilde{l}_\xi(\theta^*) \parallel^2\right] + L\gamma^2\mathbb{E}_\xi\left[\parallel \nabla\tilde{l}_\xi(\theta^*) - \lambda\hat{g}_\xi(\theta_{0:m}, \mathbf{w}_{0:m}) \parallel^2\right] \quad (37)$$

$$\leq (1 - \gamma\mu)(l(\theta_{m,t}) - l(\theta^*)) - \frac{\gamma\mu}{2}\left(1 - \gamma\frac{4L\mathcal{L}}{\mu}\right)(l(\theta_{m,t}) - l(\theta^*)) + \gamma\parallel b^{(m,t)}\parallel^2$$

$$+ L\gamma^2\mathbb{E}_\xi\left[\parallel \nabla\tilde{l}_\xi(\theta^*) - \lambda\hat{g}_\xi(\theta_{0:m}, \mathbf{w}_{0:m}) \parallel^2\right]$$

$$\tag{38}$$

$$\leq (1 - \gamma\mu)(l(\theta_{m,t}) - l(\theta^*)) + \gamma\parallel b^{(m,t)}\parallel^2 + L\gamma^2\mathbb{E}_\xi\left[\parallel \nabla\tilde{l}_\xi(\theta^*) - \lambda\hat{g}_\xi(\theta_{0:m}, \mathbf{w}_{0:m}) \parallel^2\right] \quad (39)$$

$$\leq (1 - \gamma\mu)(l(\theta_{m,t}) - l(\theta^*)) + 2\gamma\parallel \nabla l(\theta_{m,t}) - \nabla\tilde{l}(\theta_{m,t}) \parallel^2$$

$$+ 2\gamma\lambda^2\parallel \hat{g}(\theta_{0:m}, \mathbf{w}_{0:m}) \parallel^2 + L\gamma^2\mathbb{E}_\xi\left[\parallel \nabla\tilde{l}_\xi(\theta^*) - \lambda\hat{g}_\xi(\theta_{0:m}, \mathbf{w}_{0:m}) \parallel^2\right]$$

$$\tag{40}$$

$$\leq (1 - \gamma\mu)(l(\theta_{m,t}) - l(\theta^*)) + 8\gamma C^2 \cdot g^{\mathcal{E}}(\theta_m)$$

$$+ 2\gamma\lambda^2\parallel \hat{g}(\theta_{0:m}, \mathbf{w}_{0:m}) \parallel^2 + L\gamma^2\mathbb{E}_\xi\left[\parallel \nabla\tilde{l}_\xi(\theta^*) - \lambda\hat{g}_\xi(\theta_{0:m}, \mathbf{w}_{0:m}) \parallel^2\right]$$

$$\tag{41}$$

where (33) holds due to L-smoothness; (34) holds due to $g_\xi^{(m,t)} = \nabla l_\xi(\theta_{m,t}) - b_\xi^{(m,t)} = \nabla\tilde{l}_\xi(\theta_{m,t}) - \lambda\hat{g}_\xi(\theta_{0:m}, \mathbf{w}_{0:m})$; (36) holds due to $\langle a, b\rangle \leq \frac{1}{4}\parallel a \parallel^2 + \parallel b \parallel^2$; (37) holds due to the inequality $\parallel x + y \parallel^2 \leq 2 \parallel x \parallel^2 + 2 \parallel y \parallel^2$; (38) holds due to the expected smoothness assumption; (39) holds due to the condition $\gamma \leq \frac{1}{4\mathcal{L}}\frac{\mu}{L}$; (41) holds due to the law of total expectation with the random event $\mathbf{1}(Y(X) = \hat{Y}(X;\theta_m))$ and then applying the bounded support assumption.

By recursively applying the above bound to any realization of $\theta_m$, we get our desired result as follows:

$$\mathbb{E}[l(\theta_{m,t}) - l(\theta^*)|\theta_m] \leq (1-\gamma\mu)^t(l(\theta_m) - l(\theta^*)) + \frac{8C^2}{\mu} \cdot g^{\mathcal{E}}(\theta_m)$$
$$+ \frac{1}{\gamma\mu}\left(2\gamma\lambda^2 \parallel \hat{g}(\theta_{0:m}, \mathbf{w}_{0:m}) \parallel^2 + L\gamma^2\mathbb{E}_\xi\left[\parallel \nabla\tilde{l}_\xi(\theta^*) - \lambda\hat{g}_\xi(\theta_{0:m}, \mathbf{w}_{0:m}) \parallel^2\right]\right)$$
$$= (1-\gamma\mu)^t(l(\theta_m) - l(\theta^*)) + \frac{8C^2}{\mu}g^{\mathcal{E}}(\theta_m)$$
$$+ \frac{1}{\mu}\left(2\lambda^2 \parallel \hat{g}(\theta_{0:m}, \mathbf{w}_{0:m}) \parallel^2 + L\gamma\mathbb{E}_\xi\left[\parallel \nabla\tilde{l}_\xi(\theta^*) - \lambda\hat{g}_\xi(\theta_{0:m}, \mathbf{w}_{0:m}) \parallel^2\right]\right). \quad (42)$$

$\square$

**Lemma B.6** (Lemma 1 in [30]). *Let* $N(\lambda; \theta_{0:m}, \mathbf{w}_{0:m}) := \lambda^2 \parallel \hat{g}(\theta_{0:m}, \mathbf{w}_{0:m}) \parallel^2$ $+ \frac{L\gamma}{2}\mathbb{E}_\xi\left[\parallel \nabla\tilde{l}_\xi(\theta^*) - \lambda\hat{g}_\xi(\theta_{0:m}, \mathbf{w}_{0:m}) \parallel^2\right]$. *Then, for* $\lambda_m^\dagger = \frac{\mathbb{E}_\xi[\langle\nabla\tilde{l}_\xi(\theta^*), \hat{g}_\xi(\theta_{0:m}, \mathbf{w}_{0:m})\rangle]}{\mathbb{E}_\xi\|\hat{g}_\xi(\theta_{0:m}, \mathbf{w}_{0:m})\|^2 + \frac{2}{L\gamma}\|\hat{g}(\theta_{0:m}, \mathbf{w}_{0:m})\|^2}$, *we have*

$$\frac{N(\lambda_m^\dagger; \theta_{0:m}, \mathbf{w}_{0:m})}{N(0)} = 1 - \frac{\left(\mathbb{E}_\xi[\langle\nabla\tilde{l}_\xi(\theta^*), \hat{g}_\xi(\theta_{0:m}, \mathbf{w}_{0:m})\rangle]\right)^2}{\left(\mathbb{E}_\xi \parallel \hat{g}_\xi(\theta_{0:m}, \mathbf{w}_{0:m}) \parallel^2 + \frac{2}{L\gamma} \parallel \hat{g}(\theta_{0:m}, \mathbf{w}_{0:m}) \parallel^2\right)\left(\mathbb{E}_\xi \parallel \nabla\tilde{l}_\xi(\theta^*) \parallel^2\right)}.$$
$$(43)$$

# C  Additional results and discussions

## C.1  Empirical evidence for Theorem 3.1

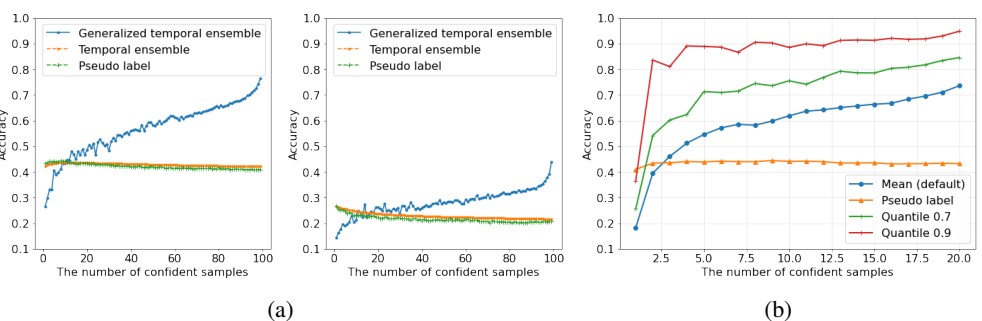

(a)
(b)

Figure 4: (a) The accuracy of the generalized temporal ensemble along with the number of confident samples under different degrees of distribution shifts. Here, the temporal ensemble is constructed by averaging all predictions over iterations. (b) On-average accuracies per the number of confident samples over iterations under different thresholding rules.

## C.2  Marginal distribution of pseudo labels over epochs

## C.3  Algorithmic design choices

Here, we examine effectiveness of our design choices–the relative thresholding for weighting scheme $w_i(x)$ and hard prediction for $p(y|x, \theta_i)$. To this end, we replace our design choices with several alternatives, keeping other features of `AnCon` the same. Then, we analyze the changes in the performance compared to the default setting of `AnCon` in four domain pairs of OfficeHome.

***Sophisticated weighting scheme for*** $w_i(x)$ We compare our relative thresholding with sophisticated weighting schemes called Entropy ($w_i(x) \propto \exp\{-H_E[f(x; \theta_i)]\}$) and Maxprob ($w_i(x) \propto \max_{k\in[K]} f_k(x; \theta_i)$). These weighting schemes aim to more precisely weight the predictions based on the uncertainty of the prediction. However, in Figure 6a, the simple relative thresholding works better than these sophisticated weighting schemes. We conjecture that the poor calibration performance of

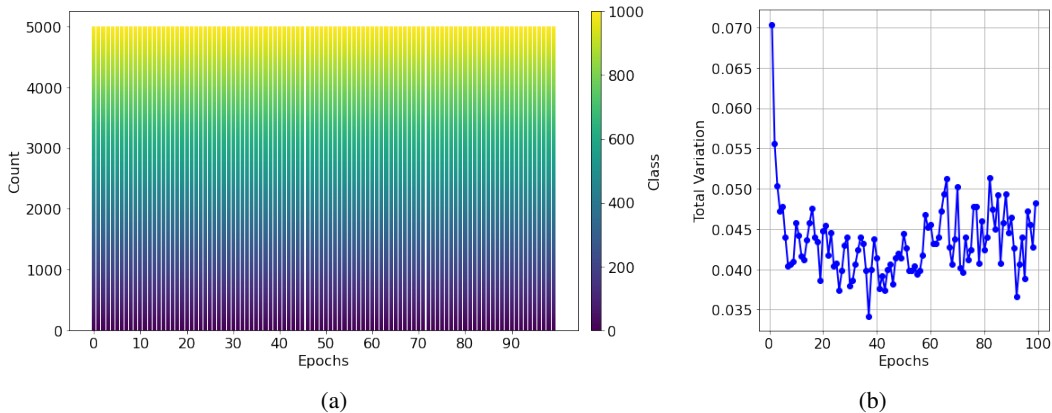

(a)                                                                (b)

Figure 5: (a) Counting the number of pseudo labels for each class with 5,000 training samples in ImageNet-C over 100 training epochs, which shows that the marginal distribution of pseudo labels barely changes during training. (b) Changes in the total variation distance of the marginal distributions of the pseudo labels for each two consecutive epochs.

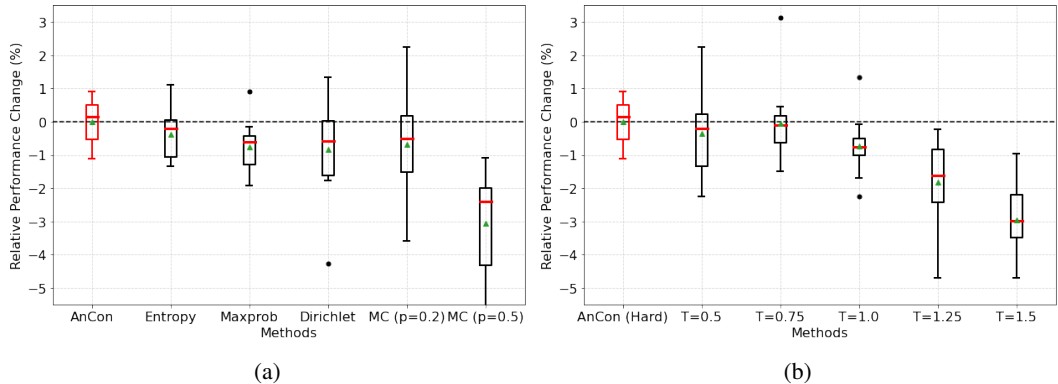

(a)                                                                (b)

Figure 6: Ablation study of (a) weighting and (b) prediction schemes.

the neural network in self-training under distribution shifts prevents the estimated uncertainty of a prediction from accurately reflecting the goodness of the prediction. Further, the ineffectiveness of these sophisticated weighting schemes persists even when we improve the calibration performance with last-layer Dirichlet [53] and Monte-Carlo Dropout [54] (see Appendix D.3 for detail). Therefore, our relative thresholding is simple yet effective in self-training under distribution shift.

***Soft prediction for*** $p(y|x, \theta_i)$ We compare hard prediction vs soft prediction with a softmax temperature parameter $T \in \{0.5, 0.75, 1.0, 1.25, 1.5\}$ (cf. Appendix D.3 for the definition). By design, soft prediction is more informative, giving values in non-leading entries. However, in Figure 6b, soft prediction turns out to underperform hard prediction for various values of $T$. We conjecture that the continuously increasing confidence in the later stage of self-training would make soft prediction ignore early-stage predictions which may be valuable to memorize. Thus, we believe that hard prediction would be more appropriate for `AnCon`.

### C.4   Limitations and future directions

Albeit `AnCon` proving its effectiveness with a fixed value of $\lambda$, Theorem 3.1 suggests that its optimal value depends on the temporal ensemble performance. Thus, future works could aim for adaptively determining $\lambda$. In addition, we observe that all self-training based methods fall short without sophisticated model designs tailored for distribution shifts [16, 31] (cf. Appendix E). In this regard, it would be valuable to rigorously understand properties of function classes that enable successful self-training under distribution shifts. Also, while `AnCon` significantly relaxes the conditions required to achieve optimality compared to LFN techniques, the on-average correct prediction condition still can be violated in challenging self-training scenarios. Given the compatibility of `AnCon` and NRC

(cf. Table 1), efficiently combining local and temporal consistencies could be a step toward further relaxing the optimality condition. Finally, we remark that the idea of selective temporal consistency in sequential decision-making scenarios could be a notable future direction of research. Extending `AnCon` to sequential decision-making scenarios presents significant challenges as they involve the fundamentally different mechanisms. In sequential decision-making, leveraging observed rewards and balancing exploitation and exploration are core aspects (e.g., constructing the upper confidence bound of the reward in the bandit problem), unlike in self-training. Therefore, the main challenges for applying `AnCon` to this setting would be defining rewards and incorporating exploration strategies into pseudo label generation.

# D  Additional details

## D.1  SFDA training configurations

Our training configuration is based on [16]. Specifically, we train ResNet-50 [29] as a source classifier, which is used as an initial point for SFDA, for 50 epochs, with minibatch stochastic gradient descent with Nesterov momentum by using the initial learning rate of 0.01, the momentum parameter of 0.9, and the minibatch size of 64 in the Office Home dataset. The learning rate is decayed by $(1 + 10 \cdot \text{current iteration number / maximum number of iterations})^{-0.75}$, and for the pre-trained layers we use a 10 times smaller learning rate. For Office-31, we change the number of epochs by 100 and other configurations are the same as those used in the setting in Office Home. Similarly, we only change the number of epochs to 10, the learning rate to 0.001, and the architecture to ResNet-101 for VisDa. We split dataset into 90% of the training set and 10% of the validation set, and the best model is chosen based on the lowest error rate on the validation set. For adapting the pre-trained model in the target domain, we use the exactly same configurations, except reducing the number of epochs to 30 for Office-31 and OfficeHome and to 15 for VisDa.

## D.2  ImageNet-C training configuration

For applying self-training, we follow the training configuration used in [31]; specifically, given ResNet-50 pretrained on ImageNet, we perform self-training without threshold by running the stochastic gradient descent optimizer with the learning rate of 0.001 and the minibatch size of 100 for 20 epochs. We remark that other techniques such as momentum and learning rate scheduling are not used.

## D.3  Further experimental details in analyses

**Definition of ECE**

$$ECE(\theta; \mathcal{D}) = \sum_{g=1}^{G} \frac{|\mathcal{G}_g^\theta|}{|\mathcal{D}|} |Acc(\mathcal{G}_g^\theta) - Conf(\mathcal{G}_g^\theta))| \tag{44}$$

where $\mathcal{G}_g^\theta := \{x \in \mathcal{D} | \frac{g}{G} \le c(x; \theta) < \frac{g+1}{G}\}$, $Acc(\mathcal{G}_g^\theta)$ is the average accuracy in $\mathcal{G}_g^\theta$, and $Conf(\mathcal{G}_g^\theta)$ is the average confidence in $\mathcal{G}_g^\theta$.

**Configuration of the sophisticated uncertainty representation methods**   We used the regularization coefficient of 0.001 for the belief matching loss [53] and 20 forward passes to compute the posterior predictive distribution for MC-dropout with dropout probabilities 0.2 and 0.5 [54].

**Definitions of model selection criteria**

- InfoMax (higher is better)

$$IM(\theta) = H(\mathbb{E}_X[f(X; \theta)]) - \mathbb{E}_X[H(f(X; \theta))] \tag{45}$$

  where $H(p) = -\sum_{k \in [K]} p_i \log p_i$ for $p \in \triangle^{K-1}$. Note that we abuse the notation $H(\cdot)$ with the definition of the cross-entropy; that is, $H(q, p) = -\sum_{k \in [K]} p_i \log q_i$ and $H(p) = -\sum_{k \in [K]} p_i \log p_i$.

Table 2: Accuracy across different domain pairs for each UDA method in OfficeHome.

| Method | Ar2Rw | Ar2Pr | Ar2Cl | Rw2Ar | Rw2Pr | Rw2Cl | Pr2Ar | Pr2Rw | Pr2Cl | Cl2Ar | Cl2Rw | Cl2Pr | Avg |
|---|---|---|---|---|---|---|---|---|---|---|---|---|---|
| CDAN | 77.48 | 71.77 | **54.23** | **74.37** | 83.60 | **61.01** | 61.39 | **79.21** | 53.95 | 62.26 | 71.08 | 69.88 | 68.35 |
| Self-training | 76.82 | 71.68 | 51.57 | 73.51 | 83.10 | 58.63 | 60.36 | 78.68 | 51.91 | 61.97 | 69.24 | 68.37 | 67.15 |
| +ELR | 77.44 | **71.91** | 47.01 | 73.96 | 83.53 | 59.98 | **61.68** | 78.88 | 51.39 | 62.88 | 70.81 | 70.33 | 67.48 |
| +AnCon | **78.22** | 71.53 | 53.59 | 74.08 | **84.03** | 60.23 | 61.23 | 78.79 | 53.81 | **62.92** | **72.04** | 70.44 | **68.41** |
| MDD | **78.54** | **75.74** | **56.43** | 73.71 | **84.30** | 60.73 | **63.70** | **80.26** | 52.99 | **63.62** | **73.10** | 72.38 | **69.24** |
| Self-training | 76.93 | 71.37 | 52.62 | 73.47 | 83.24 | 59.38 | 60.57 | 78.40 | 52.49 | 62.01 | 69.43 | 68.17 | 67.34 |
| +ELR | 77.30 | 71.93 | 47.12 | **74.00** | 83.53 | 59.84 | 62.05 | 78.75 | 50.81 | 62.30 | 70.53 | 70.29 | 67.37 |
| +AnCon | 77.67 | 71.84 | 53.24 | **74.00** | 83.96 | 60.18 | 60.86 | 79.07 | **53.38** | 62.75 | 71.63 | 70.56 | 68.26 |

- Corr-C (lower is better)

$$\text{Corr-C}(\theta) = C/(\| C \|_F /\sqrt{K}), \quad C_{ij} = \sum_{n=1}^{b} f_i(x_n; \theta) f_j(x_n; \theta) \tag{46}$$

where $C \in \mathbb{R}^{K \times K}$ and $\| \cdot \|_F$ is the Frobenius norm.
- Ent (lower is better)

$$\text{Ent}(\theta) = H(\mathbb{E}_X[f(X; \theta)]). \tag{47}$$

**Softmax with a temperature parameter**

$$p(y = j | x, \theta, T) = \phi_j(f^{-\phi}(x; \theta)/T) \tag{48}$$

where $f^{-\phi}(x; \theta)$ are the logits of the neural network such that $f_i(x; \theta) = \frac{\exp(f_i^{-\phi}(x;\theta))}{\sum_{k \in [K]} \exp(f_k^{-\phi}(x;\theta))}$ for $i \in [K]$ and $\phi(x)$ is the softmax function.

### D.4 Computational resources for experiments

In this work, we use multiple servers which consist of multiple GPUs including TITAN XP (12GB), RTX 8000 (50GB), and A100 (40GB). AnCon takes 1.16 seconds on average for each iteration.

## E Result in UDA

While SFDA does not assume any information, unlabeled target domain samples as domain shift information can be available during the source domain training process in some practical scenarios, e.g., when labeling target domain samples is costly. In this case, UDA methods are used to minimize the potential impact of distribution shifts by matching input marginal probability distributions of source and target domains. Given the prevalence of UDA scenarios in practice, we aim to show whether AnCon and other self-training methods can be effective for the UDA methods (conditional domain adversarial network (CDAN) [55] and maximum mean discrepancy (MDD) [56]) in the adaptation stage. However, as shown in Table 2, all methods decrease the performance of both CDAN and MDD for most cases, albeit the reduction rates in AnCon are smaller than self-training and ELR. We conjecture that this is because the base methods (CDAN and MDD) do not contain the sophisticated tricks, such as adding weight normalization to the final linear layer, which are used in the SFDA literature [16].

## F  Additional tables

Table 3: Benchmark results in Office-31. The numbers indicate the mean test accuracy across three repetitions with boldface for the best score.

| Method | A-D | A-W | D-A | D-W | W-A | W-D | Avg |
|---|---|---|---|---|---|---|---|
| Self-Training | 79.32 | 80.88 | 62.07 | 97.67 | 62.99 | 99.90 | 80.47 |
| +ELR | 79.02 | **81.19** | 63.70 | 97.61 | 63.63 | 99.90 | 80.84 |
| +AnCon | **82.23** | 79.94 | **63.99** | **97.80** | **64.29** | **100.00** | **81.37** |
| GCE | **86.85** | 86.16 | 64.63 | **98.62** | 62.83 | **99.90** | 83.16 |
| +ELR | **86.85** | 86.54 | 64.80 | **98.62** | 62.48 | **99.90** | 83.20 |
| +AnCon | 86.75 | **87.11** | **65.78** | 98.49 | **62.99** | **99.90** | **83.50** |
| NRC | 92.67 | 88.11 | 72.83 | 98.62 | 72.40 | 99.90 | 87.42 |
| +ELR | 92.77 | 87.92 | 72.77 | **98.68** | **72.68** | 99.90 | 87.46 |
| +AnCon | **94.28** | **91.45** | **74.49** | 98.62 | 75.56 | **100.00** | **89.07** |

Table 4: Benchmark results in OfficeHome. The numbers indicate the mean test accuracy across three repetitions with boldface for the best score.

| Method | Ar2Rw | Ar2Pr | Ar2Cl | Rw2Ar | Rw2Pr | Rw2Cl | Pr2Ar | Pr2Rw | Pr2Cl | Cl2Ar | Cl2Rw | Cl2Pr | Average |
|---|---|---|---|---|---|---|---|---|---|---|---|---|---|
| Self-training | 75.19 | 69.20 | 43.07 | 66.13 | 78.10 | 46.64 | 55.25 | 73.84 | 42.98 | 55.46 | 66.51 | 66.28 | 61.55 |
| +ELR | 75.97 | 70.47 | 45.80 | 67.00 | 78.64 | 50.01 | 55.38 | 75.42 | 44.56 | 56.61 | 69.15 | 67.38 | 63.03 |
| +AnCon | **76.13** | **70.56** | **48.06** | **67.57** | **79.27** | **51.89** | **55.62** | **76.22** | **44.86** | **58.34** | **70.23** | **68.21** | **63.91** |
| GCE | 74.65 | 69.69 | 43.46 | 68.87 | 79.25 | 49.18 | 60.01 | 75.14 | 43.78 | 57.44 | 68.85 | 65.77 | 63.00 |
| +ELR | 74.89 | 71.62 | 43.45 | 69.63 | 79.98 | 49.68 | 61.27 | 76.57 | 44.44 | 58.69 | 68.67 | 66.29 | 63.77 |
| +AnCon | **76.47** | **71.80** | **44.83** | **70.21** | **80.15** | **51.74** | **61.93** | **76.91** | **46.69** | **59.21** | **70.09** | **67.74** | **64.81** |
| NRC | 73.86 | 75.04 | 47.90 | 61.15 | 76.59 | 51.75 | 57.56 | 71.88 | 46.12 | 59.74 | 70.44 | 70.47 | 63.92 |
| +ELR | 76.82 | 76.01 | 52.19 | 66.01 | **80.40** | 55.69 | **60.82** | 74.87 | 49.62 | **63.21** | 73.56 | 72.65 | 66.89 |
| +AnCon | **79.64** | **77.11** | **53.56** | **69.47** | 80.04 | **57.30** | 59.99 | **77.37** | 50.42 | **63.21** | **75.95** | **74.97** | **67.96** |

Table 5: ECE benchmark results in OfficeHome. The numbers indicate the mean test ECE across three repetitions with boldface for the best score (lower is better).

| Method | Ar2Rw | Ar2Pr | Ar2Cl | Rw2Ar | Rw2Pr | Rw2Cl | Pr2Ar | Pr2Rw | Pr2Cl | Cl2Ar | Cl2Rw | Cl2Pr | Avg |
|---|---|---|---|---|---|---|---|---|---|---|---|---|---|
| Self-training | 13.52 | 25.10 | 42.78 | 14.97 | 15.84 | 34.16 | **18.12** | 13.98 | 35.43 | 24.67 | 18.10 | 21.97 | 23.22 |
| + ELR | 12.61 | 23.12 | 44.89 | 13.92 | 15.51 | 30.12 | 19.06 | 14.30 | 36.86 | 21.25 | 17.00 | 22.69 | 22.61 |
| + AnCon | **11.96** | **22.48** | **32.80** | **12.18** | **13.40** | **23.05** | 18.84 | **11.54** | **35.10** | **19.50** | **15.19** | **20.07** | **19.68** |
| GCE | 10.51 | 20.90 | 45.25 | 12.29 | 13.90 | 35.37 | 14.02 | 10.99 | 39.68 | 19.10 | 15.91 | 23.57 | 21.79 |
| +ELR | 13.19 | 20.74 | 46.50 | 13.84 | 15.83 | 36.35 | 15.08 | 12.66 | 40.32 | 21.09 | 20.82 | 26.27 | 23.56 |
| +AnCon | **9.63** | **18.02** | **39.20** | **10.63** | **13.54** | **28.46** | **11.24** | **9.54** | **30.86** | **16.95** | **15.79** | **21.47** | **19.52** |

Table 6: Benchmark results in VisDa. The numbers indicate the mean test accuracy across three repetitions with boldface for the best score.

| Method | Aeroplane | Bicycle | Bus | Car | Horse | Knife | Motorcycle | Person | Plant | Skateboard | Train | Truck | Avg |
|---|---|---|---|---|---|---|---|---|---|---|---|---|---|
| Source-only | 52.74 | 10.50 | 36.16 | 48.15 | 48.24 | 5.83 | 82.80 | 15.03 | 52.43 | 16.57 | 92.75 | 7.41 | 38.71 |
| Self-training | 89.69 | 53.15 | **87.08** | 57.71 | 88.34 | **99.23** | 88.39 | 77.15 | 73.38 | 16.13 | 79.86 | 0.00 | 67.77 |
| +ELR | **92.21** | 53.53 | 85.59 | **60.22** | **89.85** | 98.70 | **90.80** | 77.65 | 80.68 | 39.11 | **84.66** | 0.43 | **71.89** |
| +AnCon | 91.25 | 52.32 | 81.68 | 58.60 | 88.72 | 85.93 | 90.60 | **77.78** | 81.31 | **50.64** | 84.16 | **16.35** | 71.11 |
| GCE | 93.88 | 0.00 | **89.62** | 72.55 | 91.69 | **99.08** | 90.11 | 79.80 | 84.68 | 0.04 | 81.37 | **0.00** | 65.20 |
| +ELR | 93.50 | 18.82 | 84.50 | 68.19 | 92.05 | 98.22 | 89.80 | 82.07 | 84.04 | **6.44** | 84.89 | **0.00** | 66.90 |
| +AnCon | **95.15** | **24.00** | 86.20 | **73.26** | **93.03** | 98.36 | **91.60** | 82.72 | **88.46** | 0.83 | **87.04** | **0.00** | **68.40** |
| NRC | 30.17 | **89.50** | **83.45** | 65.20 | **95.16** | 96.05 | 83.56 | 80.73 | 89.67 | **91.10** | 86.76 | 30.68 | 74.30 |
| +ELR | 75.40 | 83.97 | 78.78 | **67.62** | 94.14 | **97.30** | **88.87** | 82.40 | **92.15** | 88.29 | **87.39** | 57.03 | 82.80 |
| +AnCon | **95.83** | 87.05 | 79.66 | 61.67 | 94.39 | 95.57 | 86.06 | 81.18 | 90.83 | 89.30 | 85.84 | **57.41** | **83.70** |

Table 7: ImageNet-C experiments with corruption intensity 1. The numbers indicate the mean test accuracy across three repetitions with boldface for the best score. We only present the first five letter for each corruption type. The full names are available in [6].

| Method | Shot | Impul | Gauss | Glass | Motio | Zoom | Snow | Frost | Fog | Brigh | Contr | Elast | Pixel | Jpeg | Speck | Satur | Spatt | Avg |
|---|---|---|---|---|---|---|---|---|---|---|---|---|---|---|---|---|---|---|
| Self-training | 68.75 | 65.48 | 69.04 | 68.78 | 71.46 | 67.17 | 67.11 | 67.60 | 71.14 | 75.00 | 72.77 | 70.83 | 72.92 | 71.24 | 70.36 | **74.24** | 73.96 | 70.46 |
| ELR | 68.76 | 65.70 | 69.14 | 69.05 | 71.54 | 67.25 | 67.32 | 67.78 | 71.19 | 74.88 | 72.87 | 70.81 | 72.92 | 71.20 | 70.52 | 74.21 | 74.00 | 70.54 |
| AnCon | **69.11** | **65.87** | **69.31** | **69.28** | **71.63** | **67.40** | **67.38** | **68.04** | **71.34** | **75.12** | **73.10** | **71.08** | **73.10** | **71.34** | **70.64** | 74.23 | **74.20** | **70.72** |

Table 8: ImageNet-C experiments with corruption intensity 2. The numbers indicate the mean test accuracy across three repetitions with boldface for the best score.

| Method | Shot | Impul | Gauss | Glass | Motio | Zoom | Snow | Frost | Fog | Brigh | Contr | Elast | Pixel | Jpeg | Speck | Satur | Spatt | Avg |
|---|---|---|---|---|---|---|---|---|---|---|---|---|---|---|---|---|---|---|
| Self-training | 63.82 | 60.36 | 64.85 | 62.21 | 68.08 | 63.33 | 59.00 | 57.62 | 69.84 | 74.09 | 71.51 | 57.85 | 72.16 | 69.03 | 67.97 | 72.79 | 70.19 | 66.16 |
| ELR | **64.06** | **60.57** | **64.99** | 63.27 | 68.27 | 63.42 | **59.26** | 58.12 | 70.00 | 74.02 | 71.54 | 58.34 | 72.24 | 69.13 | 68.21 | 72.71 | 70.32 | 66.38 |
| AnCon | **64.06** | 60.38 | 64.90 | **63.53** | **68.31** | **63.48** | 58.94 | **58.55** | **70.14** | **74.12** | **71.71** | **58.42** | **72.44** | **69.29** | **68.29** | **72.84** | **70.47** | **66.46** |

Table 9: ImageNet-C experiments with corruption intensity 3. The numbers indicate the mean test accuracy across three repetitions with boldface for the best score.

| Method | Shot | Impul | Gauss | Glass | Motio | Zoom | Snow | Frost | Fog | Brigh | Contr | Elast | Pixel | Jpeg | Speck | Satur | Spatt | Avg |
|---|---|---|---|---|---|---|---|---|---|---|---|---|---|---|---|---|---|---|
| Self-training | 56.90 | 55.37 | 56.76 | 26.37 | 57.11 | 58.38 | 59.22 | 3.92 | 67.71 | 72.93 | 66.68 | **70.65** | 69.17 | 67.21 | 60.11 | 74.26 | 66.13 | 58.17 |
| ELR | **57.33** | **55.88** | **57.21** | 45.78 | **63.00** | 60.35 | **59.61** | 43.46 | **67.86** | 72.92 | 69.20 | **70.65** | **69.35** | 67.25 | **60.33** | 74.22 | **66.33** | 62.39 |
| AnCon | 56.96 | 55.67 | 57.01 | **48.85** | 62.64 | **60.64** | 59.33 | **49.81** | 66.41 | **73.05** | **69.37** | 70.10 | 67.94 | **67.46** | 60.13 | **74.39** | 64.50 | **62.60** |

Table 10: ImageNet-C experiments with corruption intensity 4. The numbers indicate the mean test accuracy across three repetitions with boldface for the best score.

| Method | Shot | Impul | Gauss | Glass | Motio | Zoom | Snow | Frost | Fog | Brigh | Contr | Elast | Pixel | Jpeg | Speck | Satur | Spatt | Avg |
|---|---|---|---|---|---|---|---|---|---|---|---|---|---|---|---|---|---|---|
| Self-training | 14.03 | 41.54 | 26.46 | 0.61 | 54.14 | 39.06 | 52.56 | 2.83 | 65.73 | 71.21 | 41.71 | 67.08 | 64.52 | 62.36 | 53.28 | 70.06 | 61.77 | 46.41 |
| ELR | 36.10 | 43.32 | **45.07** | 32.12 | **54.45** | 56.46 | **52.98** | 41.60 | **65.90** | 71.23 | 58.27 | **67.91** | **64.78** | 62.47 | **54.45** | 70.26 | **62.41** | 55.28 |
| AnCon | **40.36** | **43.71** | 45.03 | **40.96** | 54.16 | **56.81** | 52.46 | **46.66** | 64.20 | **71.43** | 59.42 | 66.67 | 62.37 | **62.55** | 53.66 | 69.34 | 62.34 | **56.01** |

Table 11: ImageNet-C experiments with corruption intensity 5. The numbers indicate the mean test accuracy across three repetitions with boldface for the best score.

| Method | Shot | Impul | Gauss | Glass | Motio | Zoom | Snow | Frost | Fog | Brigh | Contr | Elast | Pixel | Jpeg | Speck | Satur | Spatt | Avg |
|---|---|---|---|---|---|---|---|---|---|---|---|---|---|---|---|---|---|---|
| Self-training | 0.26 | 1.72 | 1.04 | 0.57 | 0.40 | 16.11 | 34.85 | 2.63 | 60.23 | 69.14 | 0.30 | 45.69 | 61.32 | 50.50 | 33.28 | 64.70 | 51.34 | 29.06 |
| ELR | 8.00 | 14.12 | 16.00 | 4.32 | 39.59 | 50.28 | 51.14 | 33.90 | **60.54** | 69.22 | 0.37 | 56.52 | **61.71** | 55.17 | 45.75 | 64.98 | 54.47 | 40.36 |
| AnCon | **22.56** | **26.56** | **25.85** | **19.21** | **45.91** | **52.80** | **51.30** | **39.02** | 58.06 | 68.24 | **9.24** | **57.00** | 61.48 | 54.96 | 44.76 | 62.68 | 54.29 | **44.35** |

