# OpenReview forum: "Improving self-training under distribution shifts via anchored confidence with theoretical guarantees"
_NeurIPS.cc/2024/Conference — NeurIPS 2024 poster_

### Official Review · Reviewer_hP9c · 2024-07-10

**Soundness:** 3
**Presentation:** 4
**Contribution:** 3
**Rating:** 7
**Confidence:** 4

**Summary:**

This paper presents Anchored confidence (AnCon), a novel self-training algorithm to improve test-time accuracy under distribution shifts. AnCon modifies the standard self-training algorithm by adding a temporal ensemble regularization. This regularizer is constructed by the consistency of temporal ensembles weighting with their predictive confidences. This intuitively helps the model avoid the "early-learning phenomenon", i.e., the model exploiting noisy labels after initially learning clean labels. Contributions include:
1. A new algorithm for memorizing past ensemble predictions to avoid the "early-learning phenomenon", enhancing self-training performance under sequential distribution shifts.

2. In terms of theory, the authors provide a rigorous theoretical analysis for the upper bound of the test-time error of the proposed temporal ensembles by concentration inequality in Theorem 3.1. Inspired by Knowledge Distillation (KD), they also design an upper bound of the expected cross-entropy loss between the temporal ensembles and the optimal value in Theorem 3.2.

3. Extensive experiments confirm the hypothesis that the proposed algorithm can improve self-training performance regarding test-time accuracy and calibration under distribution shifts across different hyperparameter choices.

**Strengths:**

- This paper is very well-written and clear to understand the important aspects of the algorithm.
- Although the temporal ensemble weighing technique reminds me of similarities with Early learning regularization (ERL) in discount factor from past predictions, I still like the AnCon in general because of a **novel** of high-quality uncertainty estimation behavior in neural network self-training.
- Similar to ERL, AnCon is also more computationally efficient than non-parametric techniques and standard sampling-based Deep Ensembles.
- The theoretical contribution of this paper is good, yielding a solid AnCon algorithm. Specifically, under mild assumptions, Theorem 3.1 and 3.2 formally explain why AnCon works and how temporal ensemble regularizers can contribute to improving self-training under covariate shifts (details in Contribution 2).
- Experiments show a significant improvement of the standard self-training algorithm across different distribution shifts like artificial corruptions and real-world domain shifts.
- The robustness of the proposed model across different settings is also confirmed across different hyperparameter choices of coefficient regularizer $\lambda$ in Eq. 1 and discount factor $\beta$ in Eq. 3, different proportional weighting schemes for $w_m(x)$ in Eq. 3, and different soft-hard predictions for $p(y|x,\theta_i)$ in Eq. 2.

**Weaknesses:**

- Regarding the proposed algorithm: compared with the standard self-training, AnCon requires pre-defining the regularizer $\lambda$ and the discount factor for the temporal ensembles $\beta$.
 - Regarding the theory: Theorem 3.1 requires $\bar{p}(x;\mathbf{c}_{0:m}) \geq 1/2$, i.e., at least 50% accuracy on average for the temporal ensembles. This is a quite strong assumption.
 - Regarding the experiments: (1) The quantitative results in all tables are reported without the standard deviation and significant tests; (2) The improvement in calibration is shown without any theoretical evidence. Fig. 3 (a) only shows ECE without the reliability diagram, making it hard to assess the correlation between the accuracy and uncertainty of softmax’s probability; (3) Lacks of comparison with other baselines, e.g., many baselines [1] in UDA could be added to compare in Appendix C.
  - Others that are mentioned by the authors: the challenge of self-training methods under distribution shifts and the violation of average correct predictions in self-training.
- Miscellaneous: (1) Although assumptions A.1, A.2, and A.3 are mild and have been cited, I believe it is still worth discussing how they are realistic with the paper's setting; (2) Some mathematical notations are vague and perhaps need to be polished (e.g., $f(x;\theta)$ or $f_k(x;\theta)$ in L-78, how $\bar{f}(\cdot)$ in Eq. 1 connect to the Def. $\bar{f_k}(\cdot)$ in Eq. 2, $\hat{\mathbb{E}}$ in Eq. 3 seems not to be defined, the same notation $\sigma$ for different $\sigma_*^2$ in Theorem 3.2 and perhaps the softmax $\sigma_k(x;\theta_i)$ in L-339, more in Question 2); (3) I find it quite hard to follow acronyms without links, I would recommend using \usepackage[acronym]{glossaries} so readers can smoothly follow the paper; (4) Some Figure’s x-y-titles are tiny, e.g., Fig. 3 (b).

**Questions:**

1. The connection of AnCon with KD is not clear to me. The pseudo label $\hat{Y}(x;\theta)$ in self-training with AnCon and the true label  $Y(X)$ in L-163 ($Y(x)$ in L-74) are completely different because $Y(X)$ is unobservable while $\hat{Y}(x;\theta)$ is model’s prediction. So in Section 3.3, the statement in L-165-166 to connect AnCon with KD is still wrapping my head. In particular, why AnCon's pseudo-label can be considered as $Y(x)$ (perhaps the true label $Y(X)$ in L-163)? Also, I am curious about why Eq. (40) can lead to Eq. (41), i.e., how we can bound the norm of gradient loss $||\nabla l(\theta_{m,t}) - \nabla \tilde{l}(\theta_m)||^2$ by the $Err(\hat{Y};\theta_{m})$ while this upper bound related the true label $Y(X)$?
2. Some questions for clarification: is $\delta$ in L-83 $\in [0,1]$? $\sum_0^m$ or $\sum_0^{m-1}$ in Eq. 3? $\theta_k$ or $\theta_m$ in $Err_{XY}(\cdot)$ in L-184? $\mathbf{w}\_{1:m}$ or $\mathbf{w}_{0:m}$ in RHS Eq. in L-186?
3. Compared with ELR which computes the temporal ensembles by the logit vector $f(x;\theta_i)$ only in L-83, the proposed algorithm requires computing $\mathbb{E}\_{x\in X}\max_{k\in [K]}f_k(x;\theta_i)$ in Eq. 3, does this causes a higher computational demand?
4. Can you generate a reliability diagram [2] to show the overconfidence and underconfidence of methods? I think this could be easier to assess how your method can improve the model's uncertainty quality.
6. In the testing of different weighting schemes in Fig. 4 at Section 4.4, have you considered the setting of a uniform weighting for $\mathbf{w}$, i.e., set Eq. 2 to $\bar{f_k}(x;\theta_{0:m},\mathbb{w}_{0:m}):= \sum_i^m p(y=k|x;\theta_i)$? This is a simple setting but no longer temporal ensembles by discount factors, and also not standard (vanilla) self-training. I am quite curious about this result.
5. Do the authors think your proposed method can work in other sequential decision-making, e.g., bandits, RL, bayesian optimization, active learning, etc.? If yes, could you please give some comments? Otherwise, could you please raise some challenges?

References:

[1] Fang et al., Source-Free Unsupervised Domain Adaptation: A Survey, arXiv, 2023.

[2] Guo et al., On Calibration of Modern Neural Networks, ICML, 2017.

**Limitations:**

Please see my comments on the Weaknesses.

---

> ### Author Rebuttal · Authors · 2024-08-07
>
> We sincerely appreciate Reviewer hP9c's insightful suggestions and thoughtful comments. We are pleased that Reviewer hP9c acknowledges our core technical contributions to showing the effectiveness of high-quality uncertainty estimation in the temporal ensemble for improving self-training with solid theoretical and empirical results. Below, we have carefully addressed Reviewer hP9c’s valuable comments, which we believe significantly improve our work.
> (Unfortunately, due to the number of comments from Reviewer hP9c, our rebuttal addresses only Weaknesses. We will post the remaining responses to Questions in the official comment and apologize for any inconvenience.)
>
> _Note: [G#] is the reference for the global responses._
>
> **W1:** Yes, but we hold a positive view on using the provided default hyperparameter values in diverse distribution shift scenarios. Specifically, we note that the single configurations ($\lambda=0.3$ and $\beta=0.9$) work well for 105 numbers of different distribution shifts with varying difficulties. Also, the performances are shown to be not drastically affected by changes in the hyperparameter values. Finally, we kindly refer to [G4 in the global response] for more detailed explanations of our rigorous hyperparameter selection procedure and its strong practical implications.
>
>
>
>
> **W2:** We kindly refer to [G1] for the validity of the assumption of $ \bar{p}(x; c_{0:m}) $ made in Theorem 3.1. Also, we note that AnCon’s strong empirical performance can be preserved even if the condition is violated as discussed in [G3].
>
>
> **W3-1:** Thank you for pointing out the important missing details in the error bars. In the original version, we opted to omit the variance terms because the variance is not particularly large for any method (e.g., on average 0.96 for OfficeHome), which gives statistical significance at a P-value < 5%. However, in order to prevent concerns about the statistical significance, we have added P-values for each table.
>
>
> **W3-2:** Thanks for the great suggestion that can help readers to understand the uncertainty estimation behavior of AnCon. Following the Reviewer hP9c’s insightful suggestion, we have analyzed the reliability diagram. As widely known in the literature, all self-training methods turn out to be overconfident. However, as shown in Figure R5, AnCon relaxes the overconfident behavior, which can be inferred from its better ECEs in the manuscript (e.g., Table 5).
>
> **W3-3:** To clarify, AnCon aims to solve the source-free domain adaptation (SFDA) and test-time adaptation (TTA) problems, not the unsupervised domain adaptation (UDA) problem wherein the labeled source domain dataset is available during the adaptation. The unavailability of any reliable labeled samples in SFDA and TTA makes more challenging learning scenarios, which involve early learning phenomenon and model collapse under severe distribution shifts (cf. Figure 1 (a)). Therefore, the UDA methods would not be directly comparable to AnCon. Also, we note that the chosen baselines show state-of-the-art performances in general settings (cf. [G5 in the global response]).
>
> **W4:** Kindly see [G3] for the unique and significant performance gains from AnCon in challenging distribution shift scenarios, wherein the average correct predictions assumption is violated.
>
> **W5-1:** As Reviewer hP9c commented, regularity conditions in Assumptions A.1, A.2, and A.3 are mild but essential for most theoretical studies with convergence analyses. Assumptions A.1 and A.2 can trivially hold under bounded parameter values that can be guaranteed by optimizing neural networks with finite iterations under a gradient or weight clipping. Assumption A.3 holds for infinite-width neural networks, i.e., the neural tangent kernel (NTK) regime (Charles & Papailiopoulos, 2018). Given that the gradient descent training dynamics of neural networks can be well approximated by NTK (Jacot et al., 2018), the PL condition can be generally regarded as a mild assumption. We fully agree with Reviewer hP9c’s suggestion and have added these discussions to “Appendix A.1 Assumptions.”
>
> Charles, Z., & Papailiopoulos, D. (2018). Stability and generalization of learning algorithms that converge to global optima. In ICML.
> Jacot, A., Gabriel, F., & Hongler, C. (2018). Neural tangent kernel: Convergence and generalization in neural networks. In NerIPS.
>
>
> **W5-2**: Thanks for the valuable clarifying questions! $ f_k(x; \theta) $ is right; $ \bar{f}(x) := (\bar{f}_1(x), \bar{f}_2(x), \cdots, \bar{f}_K(x)) $; $ \hat{E} $ is the Monte-Carlo estimator with mini-batch samples; we have introduced $ \sigma^{2}(\bar{\theta}) $ for the variance of stochastic gradient of $\bar{f}$; finally, we introduce a new notation for the softmax as $\phi$ instead of $\sigma$ to avoid the confusion.
>
>
> **W5-3/4**: Thanks for the great suggestions. We have added links for all acronyms and increased the font size for all small figures.

---

> ### Author Response · Authors · 2024-08-07
> **Responses to questions**
>
> **Q1-1:** We inadvertently caused confusion to Reviewer hP9c on the connection between AnCon and knowledge distillation (KD). To be clear, we do not intend to treat the pseudo label $ \hat{Y} $ as the true label $ Y $ since this reduces the theoretical rigor of our work. Indeed, we clearly state the bias coming from pseudo labels in the optimality gap in Theorem 3.2. Also, we state that “the gradient is biased due to the usage of pseudo labels and the generalized temporal ensemble” in Line 168-169. The purpose of this connection is to show how AnCon can reduce the variance of stochastic gradients in the self-training scenario through the partial variance reduction theory by interpreting our generalized temporal ensemble as the teacher network. In order to avoid any confusion, we have changed Lines 165-169 as follows:
>
> “””
> … , where the generalized temporal ensemble is the teacher network $ f^{(t)} $ with a notable difference that the pseudo label $ \hat{Y} $ is used instead of the true label $ Y $ which requires careful analysis for studying the optimality. The purpose of this connection is to perform a convergence analysis of AnCon by modifying the partial variance reduction theory [31], as given below. We note that the usage of pseudo labels, instead of the true label, and the generalized temporal ensemble result in an inherently biased gradient estimator. Therefore, the convergence analysis requires special treatments for handling the biased gradient, unlike typical supervised learning settings in self-distillation literature.
> “””
>
>
> **Q1-2:** It can be derived by using the law of total expectation with the random event $ 1(Y(X) = \hat{Y}(X)) $ and then applying the bounded support assumption, which we have added to Line 678.
>
>
> **Q2-1:** If this refers to Line 125 in Eq 3, yes due to the recursion.
>
> **Q2-2:** It is \sum_{0}^{m}, which means the current average confidence is taken into account because the exponential moving average is introduced to avoid the computation of the expectation with respect to all samples.
>
> **Q2-3:** Thank you so much for your careful review and pointing out the type. It is $\theta_m$ because $Err_{XY}(\hat{Y}; \theta_{m,0})$ is an error rate of pseudo label under parameter $\theta_m$.
>
> **Q2-4:** Thanks again for pointing out the typo. This is $ w_{0:m} $ because the generalized temporal ensemble includes the initial model prediction. (After very careful review, we have found other typos in Line 178, Line 179, Line 674, which we corrected).
>
>
>
> **Q3:** AnCon has almost the same computational complexity as ELR, and the confusion of Reviewer hP9c would be due to our undefined notation of $ \hat{E}[c(X; \theta_i)] $ which is a Monte-Carlo approximation of $ E[c(X; \theta_i)] $ with mini-batch samples $ \xi $. Specifically, Eq (2) for storing previous predictions is the same as with storing the previous logit vector in ELR ($ K \times N $ numbers). Also, Eq (3) can be efficiently implemented by using softmax output values during the training process which are already computed for computing the self-training loss. Thus, Eq. (3) requires storing only a single number with the computational cost of adding $ B $ numbers. In order to prevent the confusion, we have added the above discussion in Line 131.
>
>
> **Q4:** Kindly see the response to the weakness.
>
>
> **Q5:** As per Reviewer’s insightful comment, we have tested the undiscounted vanilla temporal ensemble (UVTE) method, which results in 4.2\% of reduction of accuracy on average. Intuitively, UVTE does not work well because memorizing all predictions, which involve highly inaccurate ones, can result in a low-quality temporal ensemble. Indeed, this is well illustrated in Figure R3 where the UVTE would not take advantage of gathering more samples due to inclusion of more incorrect predictions compared to AnCon. We believe this new result further highlights the importance of uncertainty awareness for the temporal ensemble in self-training, which is consistent with our theoretical results in Theorems 3.1 and 3.2.
>
>
> **Q6:** Thank you for the insightful question that made us think about the non-trivial extension of AnCon. Extending AnCon to sequential decision-making scenarios presents significant challenges as they involve the fundamentally different mechanisms. In sequential decision-making, leveraging observed rewards and balancing exploitation and exploration are core aspects (e.g., constructing the upper confidence bound of the reward in the bandit problem), unlike in self-training. Therefore, the main challenges for applying AnCon to this setting would be defining rewards and incorporating exploration strategies into pseudo label generation. As it holds great significance, we mark this extension as an important future direction.

---

> > ### Comment · Reviewer_hP9c · 2024-08-11
> >
> > I thank the authors for the detailed rebuttal, especially for clarification for Q1 and new results in Q4-5. I keep my original rating for this paper. Since other reviewers also need clarification on the typos, I hope the authors can carefully revise the notation in the final version of the paper. Good luck!

---

> > > ### Author Response · Authors · 2024-08-11
> > >
> > > Thank you for your thoughtful review and for acknowledging the clarifications and new results we provided. We fully understand the importance of clear notation, which we have significantly improved thanks to the reviewers’ suggestions. We will carefully revise the notation and correct any typos in the final version of the paper. Thank you again for your insightful comments throughout the review process!

---

### Official Review · Reviewer_qtUp · 2024-07-12

**Soundness:** 3
**Presentation:** 3
**Contribution:** 3
**Rating:** 6
**Confidence:** 3

**Summary:**

The paper aims to improve self-training, a common technique used for learning on unlabeled data (with iteratively generated pseudo labels), when it is applied to scenarios with distribution shifts. The proposed approach, Anchored Confidence (AnCon), essentially applies label smoothing on the pseudo labels with the ensemble prediction of the models obtained from previous iterations (temporal ensemble) to achieve better temporal consistency and improve the quality of the pseudo labels. Theoretical analysis is provided on the benefits of using temporal ensemble and the connection to ensemble knowledge distillation in the linear case. Empirically, when AnCon is shown both **effective** and **stable** when combined with a few orthogonal baselines such as Self-training and GCE under settings such as unsupervised domain adaptation (Sec. 4.1), robustness against label corruption (Sec. 4.2), calibration, etc. (Sec. 4.3).

**Strengths:**

1. The main paper is well-written and pleasant to read.
2. The paper effectively modifies and adapts Early Learning Regularization (ELR) to the setting of self-training under distribution shifts with both theoretical justification and empirical evidence. The theoretical derivations are generally rigorous with assumptions clearly stated.
3. The connection to self-distillation helps to better understand the approach.
4. The empirical and ablation studies are comprehensive and clearly discuss a few critical criteria such as model selection, hyperparameter, and calibration.

**Weaknesses:**

1. Empirically, the approach improves over various baselines when being added, but what about the existing SoTA baselines for unsupervised domain adaptations, which seem relevant to this work. Besides, it seems rather odd to only report the results in Sec. 4 with non-optimal $\lambda=0.3$, as the sensitivity analysis in Figure 2(a) clearly prefers much larger $\lambda$.
2. A critical assumption in the theorems is that the teacher (temporal ensemble) has a sufficiently good performance. It will be helpful to justify this with more empirical evidence.
3. Non-trivial typos: In line 149, $x$ is already used in the main theorem, so may want to change $x$ to $z$ to avoid confusion, and this issue also occurred on 643 with $m$, which might have already caused *major* problems in proof (see below in Question 3)
4. Minor typos: Line 667, Proof of Lemma A.4, Eq. (31), $k$ should be written as $mq$; Some references from NeurIPS 2024 should be 2023.
5. There might be some mistakes in the theorem proofs. See below in Questions 3 and 4.

**Questions:**

1. The pseudo label $\tilde{Y}$ is actually also part of the smoothing vector from $\bar{f}$. I wonder how necessary it is to separate $\tilde{Y}$ out as its purpose is to increase the weight of the most recent model, or whether there is a more unified notion to describe the proposed approach with just $\bar{f}$.
2. In Theorem 3.1, the random events of model predictions from different iterations $i, j$ are assumed to be independent. How realistic is the assumption given that the model is trained based on the labels given by the previous iterations?
3. Throughout the paper, $m$ is already defined as the number of models, but the proof of Theorem 3.1 assigns $m = Q(x;c_{0;m})/2$, which is the number of sufficiently confident predictions for $x$ made by the $m$ models. I don’t think it’s appropriate to assign $m$ something else here as it is already defined. If you fix the notation of $m$ (of $q \cdot m$) in Eq. (29) and Eq. (31), Lemma A.4, the bound may look a bit more complex, because the $m$ in Eq. (30) cannot be assigned to a different value. Please clarify.
4. Moreover, in line 152, the paper claims $\bar{f}$ is asymptotically correct, but it is unclear to me if the probability on the L.H.S. of Eq. (6) still goes to 0 when $m \rightarrow \infty$ after fixing the aforementioned problem with $m$. Please clarify.
5. Theorem 3.1 studies the asymptotic correctness of the temporal ensemble with large $m$. The experiments mostly show the AnCon under 20 epochs. How does the performance change when more epochs are added? Is it still consistent with the theoretical implications?

**Limitations:**

The authors have discussed the limitations in Section 6.

---

> ### Author Rebuttal · Authors · 2024-08-07
>
> We sincerely thank Reviewer qtUp for the insightful suggestions and constructive comments, which have significantly improved the clarity of our paper. We are glad that the reviewer enjoyed reading our paper and acknowledged our principled improvements over the existing temporal consistency methods with theoretical guarantees and attractive properties. In our response below, we have addressed the questions and comments.
> _Note: [G#] is the reference for the global responses._
>
> **W1-1:** To clarify, AnCon aims to solve the source-free domain adaptation (SFDA) and test-time adaptation (TTA) problems, not the unsupervised domain adaptation (UDA) problem wherein the labeled source domain dataset is available during the adaptation. The unavailability of any reliable labeled samples in SFDA and TTA makes more challenging learning scenarios, which involve early learning phenomenon and model collapse under severe distribution shifts (cf. Figure 1 (a)). Therefore, the UDA methods would not be directly comparable to AnCon. Also, we note that the chosen baselines show state-of-the-art performances in general settings (cf. [G5 in the global response]).
>
>
> **W1-2:** Our suboptimal choices of hyperparameters are due to our rigorous and practical hyperparameter choice setting, and we kindly refer to [G4] for detailed explanation on the hyperparameter selection procedure and its strong practical implication.
>
>
> **W2:** Thanks for pointing out the significance of the assumption made in Theorem 3.2, which asserts that "AnCon is at least better than vanilla self-training." However, the assumption is required only for deriving the final bound in Eq (7) to gain more insights about the impact of the generalized temporal ensemble's quality on the suboptimality of self-training. Rather, the core message of Theorem 3.2 that “AnCon is at least better performance than the vanilla self-training” holds _without_ this assumption. Specifically, in Eq. (13) on page 15, $ N(\lambda^\dagger) / N(0) $ corresponds to the ratio between the last term of Eq. (12) under AnCon and the vanilla self-training. Crucially, this ratio is less than or equal to 1, supporting the central message without the assumption.
>
> Thanks to the Reviewer qtUp’s insightful question, we have separated the statement of Theorem 3.2 into two parts and modified the discussion in Lines 187-190 (Kindly see the comment below for the final form) to avoid the initial confusion and emphasize that AnCon guarantees at least better performance than the vanilla self-training can hold without such assumption.
>
>
>
>
> **W3:** Thanks for pointing out the notation mistake. Since the argument of the function $ \xi $ is an arbitrary scalar, we have changed $ x $ to $ z $ in Line 149. We have changed the index notation of $ m $ in the Lemma A.4. to $ o $. These notational changes do not invalidate the proof. Regarding the confusion from the index notation, kindly see the response to Q3.
>
>
> **W4:** Thanks for pointing out the minor typos! We have changed $ k $ by $ m \cdot q $ and fixed typos for the references.
>
>
> **W5:** Kindly see the responses to Q3 and Q4.
>
>
> **Q1:** Thanks for suggesting to unify the notation of the generalized temporal ensemble $ \bar{f} $ and the pseudo label $ \hat{Y} $ that might be clearer than the current notation. Despite careful thoughts, unfortunately, we do not see any feasible way to unify the notation without sacrificing the insights and clarity that our current notation can provide. Specifically, our separate notation enables clear explanations of the selective temporal consistency (cf. Eq (2)) that is only involved in $ \bar{f} $. Also, the notation can intuitively explain the role of $\bar{f}$ that regularizes the pseudo label $ \hat{Y} $ through label smoothing, which makes the discussions in Lines 105-115 and the connection between AnCon and knowledge distillation clearer. Finally, there are points that require discussion solely on either of them (e.g., when referring to the accuracy of pseudo labels in Line 145 and Line 184; when discussing the quality of the generalized temporal ensemble in Theorem 3.1). Given the above reasons, we hope that Reviewer qtUp could agree with the advantage of separating notation for the pseudo label and the generalized temporal ensemble.
>
>
> **Q2:** Thanks for highlighting the validity of the assumption in Theorem 3.1. Here, we emphasize the important aspects of the assumption: the assumption is based on the “conditional” independence between temporal correctness given that the current prediction on a sample is relatively confident. This is grounded in the strong correlation between accuracy and confidence across a wide range of scenarios. Another way to understand this assumption is that the previous history of predictions is summarized in the random event whether the prediction is confident or not. We further remark that the validity of this assumption could be reinforced by strong empirical support for Eq. (6), as shown in Figure R3.
>
>
> **Q3:** Thanks for pointing out the unclear notation for the supporting lemma. The number of random Bernoulli samples $m$ in the Lemma A.4 (which we change to $o$) is not used in the other parts. Therefore, all other parts remain the same, except Line 624 which refers to this lemma is changed to “... due to Lemma A.4 with $ p_i = P(Y(x) = \hat{Y}(x; \theta_i)) $, $ o = Q(x; c_{0:m}) $, and $ q= 1/2 $.”
>
>
> **Q4:** We kindly refer to [G2] for the clarification of the notion of asymptotic convergence. Also, we note that [G2] presents the important implication of Eq. (6) in the non-asymptotic region, i.e., monotonic improvement of quality of the generalized temporal ensemble, with strong empirical evidence.
>
>
> **Q5:** As explained in [G2], we kindly refer to Figure R3 that shows monotonic improvements over 100 epochs of quality of the generalized temporal ensemble under diverse distribution shifts scenarios, being consistent with Theorem 3.1.

---

> > ### Comment · Reviewer_qtUp · 2024-08-09
> > **Reviewer Response**
> >
> > Thank you for the responses for addressing my concerns including the asymptotic convergence w.r.t. $Q$ and the choice of hyperparameters.
> >
> > For W3 and Q3: In Eq. (30), $m$ is the number of models. In Eq. (31), you set $t = \log \frac{k}{m\bar{p}}$, is this $m$ the number of model or the new $o$? Let’s suppose $m$ is changed to $o$, such that $t=\log \frac{q}{\bar{p}}$ here, then the bound in Eq. (31) may look like $\exp(oq-m\bar{p}-oq\log\frac{q}{\bar{p}})$ with both $m$ and $o$ inside. I think this should affect the bound in Theorem 3.1 as the number of models $m$ would appear. It’s better for the authors to show a more complete derivation after fixing the notations.

---

> ### Author Response · Authors · 2024-08-07
> **Detailed changes as per the response to W2**
>
> “””
> Assume $ l(\theta) $ satisfies $ L $-smoothness, $ \mathcal{L} $-expected smoothness, and $\mu$-Polyak-Lojasiewicz (PL) condition (cf. Assumptions A.1, A.2, and A.3).
> For $\gamma \leq \tfrac{\mu}{4 \mathcal{L} \cdot L}$ and a carefully chosen $\lambda$ (cf. Lemma A.6), it holds that
> 			$$ Eq. (12) $$
>
> Further, if $\mathbf{w}_{0:m}$ such that {...}, i.e., the teacher has a sufficiently good performance, the optimality gap becomes
>
> $$ Eq. (7) $$
>
> where $l^*$ {...} .
>
> “”” (here we could only refer equation number and use {...} to represent the repetitive equations since the MathJax cannot display the equations properly.)
>
>
> Modified discussion in Lines 187-190:
> “””
> In Theorem 3.2, we remark that the vanilla self-training ($ \lambda=0 $) results in the maximum value of the last term in Eq. (12), which is the neighborhood size of the stochastic gradient descent, while having the same first two terms. In other words, the result suggests that AnCon is at least better than vanilla self-training under the mild regularity conditions. Furthermore, under the assumption of a sufficiently good performance temporal ensemble, we gain valuable insights on the design of $ w_{0:m} $: aiming for …
> “””

---

> ### Author Response · Authors · 2024-08-09
> **Further clarification**
>
> Thanks for reading the authors' rebuttal and asking a further clarifying question.
>
> The confusion of Reviewer qtUp is noted. In our changed notation, all $ m $ in Lemma A.4. are changed to $ o $, which is the number of random Bernoulli samples. We remark that Lemma A.4 is a supporting lemma, which means that the semantics of notations are independent of other parts of the paper (i.e., none of the quantities in Lemma A.4, including Eq. (30), is interpreted as the number of models).
>
>
> Therefore, all $ m $ notations in Lemma A.4 are changed to $ o $. Then, the right-hand side of Eq. (31) is $ o q - o \bar{p} - o q \log \frac{q}{\bar{p}} $, which is obtained by $M(t) = exp( o \bar{p} (exp(t) - 1))$ and $ t = log \frac{q}{\bar{p}} $.
>
>
> From the above, Eq. (29) is $ p(S_o \leq q o) \leq exp(o (q - \bar{p} - q \log \frac{q}{\bar{p}}) ) $. Therefore, applying this inequality to Theorem 3.1 with $ p_i  = p(Y(x) = \hat{Y}(x; \theta_i)) $, $ o = Q(x; c_{0:m}) $, and $ q = 1 / 2 $ proves the last inequality in Eq. (11).
>
> We hope this clarifies why there is no mistake in the proof and that the confusion arose from the same notation used in the supporting lemma and other parts of the paper. We sincerely appreciate Reviewer qtUp's thorough review and attention to detail.

---

> > ### Comment · Reviewer_qtUp · 2024-08-10
> > **Reviewer Response**
> >
> > Thanks for the clarification. I apologize for mistakenly thinking $m$ for $S_m$ should not be updated to $o$. Now everything is clear! The other two reviewers are also generally positive about the contribution of this work, with minor similar concerns about the assumption of $\bar{p}(x;c_{0:m}) \geq 1/2$. Considering the soundness of theoretical analysis and the empirical advantages, I believe this paper will greatly contribute to the research on self-training and test-time adaptation. I have raised my score to 6 for acceptance :)

---

> > > ### Author Response · Authors · 2024-08-11
> > >
> > > Thank you for taking the time to revisit our explanation. We're glad that the clarification resolved the initial confusion regarding the notation. Also, we appreciate your positive evaluation of our work and your consideration of the theoretical analyses and empirical results. Your insights have been so much helpful in improving the clarity and rigor of our paper!

---

### Official Review · Reviewer_fUqB · 2024-07-12

**Soundness:** 3
**Presentation:** 3
**Contribution:** 3
**Rating:** 7
**Confidence:** 4

**Summary:**

the authors propose a novel approach to enhance self-training for test-time adaptation (TTA) or source-free domain adaptation (SFDA) in neural networks facing distribution shifts. The core idea revolves around a method called Anchored Confidence (AnCon), which uses temporal ensembles and label smoothing to improve the accuracy and robustness of pseudo labels generated during self-training. This method aims to address the challenge of filtering incorrect pseudo labels, a common issue under distribution shifts, without incurring significant computational overhead. Theoretical guarantees and extensive experiments validate the efficacy of AnCon in diverse distribution shift scenarios.

**Strengths:**

* the paper provides rigorous theoretical analyses to support the proposed method. It shows that the generalized temporal ensemble with prediction confidences is asymptotically correct and that label smoothing can reduce the optimality gap.
* The method does not require additional forward passes or neighborhood searches, making it computationally efficient compared to existing techniques.
* experimental results demonstrate improvements in self-training performance across distribution shift scenarios.

**Weaknesses:**

* The novelty of the paper is limited! (*The integration of temporal consistency and ensembles for self-training under distribution shifts)
* Although the method shows good empirical results on small datasets, there may be scenarios or datasets where the performance gains are less pronounced, which are not extensively discussed.
* The success of the method is somewhat dependent on the quality of the initial model parameter $\theta_0$. If the initial model is significantly biased or underperforming, the improvements might be limited.

**Questions:**

* The paper claims that the generalized temporal ensemble with prediction confidences is asymptotically correct. Can you provide a detailed proof or further elaboration on the conditions under which this asymptotic correctness holds? Specifically, what assumptions about the data distribution or model convergence are necessary to guarantee this property?

* The theoretical analysis suggests that label smoothing can reduce the optimality gap in self-training. Could you explain the underlying mechanism of how label smoothing achieves this reduction? Additionally, what are the theoretical bounds on the optimality gap with and without label smoothing, and how do these bounds depend on the hyperparameters of the smoothing technique?

* How does AnCon perform on highly imbalanced datasets or with extreme distribution shifts?
* What are the specific hyperparameter settings used in the experiments, and how sensitive is the method to these settings?
* Can the proposed method be integrated with other advanced TTA or SFDA techniques, and what would be the potential benefits or drawbacks?
* How does the method handle concept shifts, where p(Y|X) changes between the training and test distributions?
* are there any specific types of neural network architectures or tasks (e.g., vision, NLP) where AnCon performs exceptionally well or poorly?

**Limitations:**

Please see the questions and the weaknesses.
My current decision on this paper is borderline (reject/accept). I look forward to the authors' rebuttal to address the following questions and concerns, which will help in making a final decision.

---

> ### Author Rebuttal · Authors · 2024-08-07
>
> We sincerely thank Reviewer fUqB for the valuable suggestions and comments. We are glad that the reviewer acknowledges our rigorous theoretical analyses, efficient algorithm development, and effective performances of AnCon under different distribution shifts scenarios. We believe that addressing the valuable comments further improves our paper, which we hope can effectively address any concerns of Reviewer fUqB.
> * [G#] is the reference for the global responses.
> (Unfortunately, due to the number of comments from Reviewer fUqB, our rebuttal addresses parts of responses. We will post the remaining responses in the official comment and apologize for any inconvenience.)
>
>
> **W1:** Thank you for highlighting the novelty aspect of our paper. As Reviewer fUqB noted, promoting temporal consistency is a well-established framework in self-training algorithms. However, our contribution lies in demonstrating that "uncertainty-awareness" enhances the effectiveness of temporal consistency under distribution shifts. Specifically, Theorem 3.1 shows that uncertainty awareness improves the quality of temporal ensembles proportional to the number of temporally confident predictions. Additionally, Theorem 3.2 and Corollary 3.2.1 illustrate that "AnCon" reduces the optimality gap of self-training, which can be further improved by a high-quality, uncertainty-aware teacher. Crucially, such high-quality teachers are driven by our simple thresholding rule (Eq. (3)) with theoretical guarantees (Theorem 3.1). The significant theoretical contribution has not been studied and would not be possible by just incorporating the temporal consistency through ensemble as shown in Figure R6. In order to help readers to conceptualize the core theoretical contributions of the paper, we have the following discussion to Line 199.
>
> ‘’’
> Furthermore, we highlight the significance of AnCon’s uncertainty awareness for improving effectiveness of temporal consistency in self-training. Specifically, from the result of Theorem 3.1, $g^{KL}(w_{0:m})$, thereby the optimality gap of the self-training, can be further reduced by collecting more confident samples, while guaranteeing the condition $\bar{p}(c_{0:m}) \geq 0.5$. Therefore, it is important to collect high-quality predictions for matching the condition, which would be hard to satisfy under other temporal ensemble mechanisms (cf. Figure R3).
> ‘’’
>
> **W2:** Kindly see [G3 in the global reference] that explains the unique and significant performance gains from AnCon in challenging distribution shifts scenarios.
>
> **W3:** Thank you for pointing out concerns regarding the dependency on the initial model parameter. As Reviewer fUqB conjectured, we have rigorously shown the impact of initial model performance on the sub-optimality of AnCon and the vanilla self-training method (cf. $l(\theta_0) - l^*$ and $Err(\hat{Y}; \theta_0)$ in Corollary 3.2.1). Crucially, in Corollary 3.2.1, both AnCon and the self-training method can improve by reducing the initial optimality gap  $l(\theta_0) - l^*$ as the number of inner and outer iterations $T \cdot (m+1)$ increases, which is why we need “adaptation” if the performance deteriorates under severe distribution shifts. Finally, we kindly refer to [G3] for the discussion of empirical results supporting this argument.
>
> **Q1:** We gently remind Reviewer fUqB that the asymptotic convergence on inputs on which $ f $ produces infinitely many relatively confident predictions is rigorously proven in Theorem 3.1 with explicitly stated assumptions. We also kindly refer to [G1] for the discussion about validity of assumptions and [G2] for the empirical evidence.
>
> **Q2:** This is actually rigorously proven in Theorem 3.2, a core technical contribution of our work, as discussed in Lines 187-199. Specifically, “the underlying mechanism of how label smoothing achieves this reduction” is through connecting AnCon with the self-distillation. Crucially, this connection enables the adoption of the partial variance reduction theory, which can show that the generalized temporal ensemble reduces the neighborhood size of SGD.
> Our convergence analysis provides the optimality gap in Eq. (7), with the maximum gap occurring in self-training without label smoothing (cf. Line 187-188). This demonstrates that AnCon is at least better than vanilla self-training (cf. Line 189). Additionally, AnCon further improves the optimality gap by reducing $g^{KL}(w_{0:m})$, which explains the effectiveness of selective temporal consistency.
>
> **Q3:** The impacts of severe distribution shifts typically result in a poorly performing initial model, which can significantly reduce the effectiveness of self-training methods like AnCon. However, as discussed in our response to W3, AnCon successfully improves poorly performing initial model parameters under severe distribution shifts, unlike other self-training methods. Further, following Reviewer fUqB’s suggestion, we performed additional experiments in the highly class imbalance scenario. This setting is significant as effective methods like oversampling are not applicable in self-training, yet they largely impact self-training performance. In the new experiments, we compare AnCon with ELR and vanilla self-training under a heavy-tailed class imbalance with an imbalance ratio of up to 65 times. As shown in Figure R2, AnCon's effectiveness is well-preserved under severe class imbalance scenarios. In summary, we believe AnCon would show promising performances in multiple challenging self-training scenarios.
>
> **Q4:** Kindly see [G4] for the detailed explanation of our rigorous hyperparameter selection procedure and its significant practical implications. We also kindly refer to “Section 4.3.2. Robustness to the choice of hyperparameters” which shows the stable performances of AnCon even under arbitrary choices of hyperparameters.

---

> ### Author Response · Authors · 2024-08-07
> **Responses to Q5 and Q6**
>
> **Q5:** We kindly refer to “Section 4.1” for the integration of AnCon with state-of-the-art SFDA technique (NRC) and TTA technique (GCE). We also kindly refer to [G5 in the global response; GLOBAL SOTA] for conceptualizing the state-of-the-art performance levels of considered baselines.
> Regarding potential benefits, AnCon differs from state-of-the-art methods in handling noisy pseudo labels under distribution shifts. Specifically, NRC filters incorrect predictions based on local consistency, while AnCon uses temporal consistency. Combining NRC and AnCon leverages pseudo labels that are both locally and temporally consistent, resulting in significant performance improvements over NRC or Self-Training + AnCon (cf. Table 1).
> In addition, GCE reduces the impact of wrong pseudo labels rather than finding them. Applying GCE to AnCon minimizes the effects of potentially wrong but temporally consistent pseudo labels, which can be implied by the performance of GCE + AnCon compared to GCE or Self-Training + AnCon (see Table 1).
> In summary, AnCon can complement existing state-of-the-art methods by handling noisy pseudo labels fundamentally differently.
>
>
> **Q6:** We kindly remind Reviewer fUqB that the concept shift is out of the scope of our work as stated in Line 71. Also, we note that addressing concept shifts in the self-training scenario without labels is questionable, as even detecting such shifts without feedback on predictions through shifted labels in a principled manner is impossible (e.g., Lu et al., 2018).
>
> - Lu et al. (2018). Learning under concept drift: A review. IEEE T-KDE.
>
> **Q7:** We appreciate the curiosity of Reviewer fUqB about extending AnCon to various neural network architectures and tasks. Indeed, our theory and algorithm for AnCon are broadly applicable as AnCon is independent of specific neural network architectures or dataset structures, unlike methods that rely on batch normalization statistics or random image augmentations. At the same time, being agnostic to architectures and data structures make it inherently hard to make rigorous theoretical statements on the extensibility. Therefore, affirmative conclusions about the extension of AnCon would require large-scale comprehensive experiments with careful design of explorative strategies, which is beyond the scope of this paper.
> Moreover, even if AnCon performs suboptimally in certain architectures or tasks, this does not invalidate (1) our theoretical contribution of showing how uncertainty-aware temporal consistency improves self-training under distribution shifts and (2) our algorithmic contribution of designing an effective temporal ensemble mechanism without necessitating computationally heavy processes such as additional forward passes or neighborhood searches.
> In summary, while we acknowledge the potential benefits of further explorations, we believe our current focus provides a solid foundation for understanding AnCon's impact and effectiveness.

---

> > ### Comment · Reviewer_fUqB · 2024-08-12
> > **Official Comment by Reviewer fUqB**
> >
> > I have carefully reviewed the feedback from other reviewers, considered the author’s rebuttal, and followed the ensuing discussion. I appreciate the authors' thorough responses, particularly their clarifications on W1 and answering my questions.
> >
> > Assuming that the insights from these discussions will be included in the final paper, I recommend the paper for acceptance as it provides interesting insights and has the potential to contribute to the ML community and I will raise my score from 5 to 7.

---

> > > ### Author Response · Authors · 2024-08-12
> > >
> > > Thank you for your careful review and for taking the time to consider our rebuttal and the feedback from other reviewers. We greatly appreciate your acknowledgement of our efforts to clarify W1 and address your questions. We will ensure that the insights from these valuable discussions are thoroughly incorporated into the final version of the paper.
> > >
> > > Thank you once again for your thoughtful evaluation and support!

---

### Author Rebuttal · Authors · 2024-08-07

**G1:** On the validity of the on-average assumption
In Theorem 3.1, we made an assumption that $ \bar{p}(x; c_{0:m}) > 0.5 $, which we argue is not strong because of its dependence on the confidence thresholds $ c_{0:m} $. Specifically, $ \bar{p} $ is measured only for relatively confident predictions (cf. Line 144-145). This means that the assumption can hold by controlling the confidence threshold (cf. $ c_{0:m} $ in Theorem 3.1) at the expense of loosening the upper bound in Eq. (6) as mentioned in Line 156-157. For a concrete example, we have added Figure R1 that explains both (1) the average accuracy can be higher than 50% in the challenging setting where the pseudo label accuracy is below 50% and (2) $ \bar{p} $ can be further increased by increasing the confidence threshold (e.g., select 90th-quantile).

**G2:** Further elaborations of Theorem 3.1
In Theorem 3.1, the convergence holds for the asymptotic region of the number of confident predictions over iterations $ Q(x; c_{0:m}) $, not the number of iterations $ m $; that is, the inputs on which the neural network produce relatively confident predictions infinitely many times as the number of iterations goes to infinity.

Further, we remark that the bound monotonically decreases as $ Q(x; c_{0:m}) $ increases. This means that the generalized temporal ensemble can provide high-quality learning signals for self-training in non-asymptotic regions through the “uncertainty-aware” temporal consistency that helps to satisfy the condition $ \bar{p}(x; c_{0:m}) > 0.5 $. Specifically, as shown in our new figures (cf. Figure R3), our generalized temporal ensemble’s accuracy tends to increase by a margin significantly as the number of confident samples increases, being consistent with our theory. We note that this monotonic improvement would not be the case for the temporal ensemble without uncertainty-awareness and vanilla self-training, which highlights the importance of ‘selectivity’ for constructing the temporal ensemble.

To clarify the notion of asymptotic convergence and monotonic improvement, we have changed Line 151-152 as follows:

“””
The result says that as long as the expected accuracy for relatively confident predictions is at least 50% on average over iterations, $ \bar{f}(x; \theta_{0:m}, w_{0:m}) $ is asymptotically correct on $ x $ such that $ Q(x; c_{0:m}) \rightarrow \infty $ as $ m \rightarrow \infty $. Besides, we remark that the error rate of the generalized temporal ensemble monotonically decreases as $ Q(x; c_{0:m}) $ increases if the iteration on-average accuracy condition holds.
“””

Also, in Line 143, we have added “for samples where the neural network tends to be relatively confident during the self-training” after “asymptotically correct.”


**G3:** On performance of AnCon under severe distribution shifts
Performance gains over a poorly performing initial parameter through self-training are shown in our extensive experiments. Specifically, in challenging scenarios where the initial model trained on the source domain significantly deteriorates, AnCon significantly improves performance, unlike vanilla self-training. This aligns with our theoretical results in Theorems 3.1 and 3.2. For example, in VisDa-2017 (Table 6), while the initial model accuracy is 38.71%, AnCon and self-training improve it to 71.11% and 67.77%, respectively. Additionally, for Shot, Impulse, and Gaussian corruptions with the most extreme shift intensity of 5, where the initial model achieves accuracies of (3.04%, 1.76%, 2.12%), AnCon achieves (22.56%, 26.56%, 25.85%) (cf. Table 11). This striking improvement, compared to vanilla self-training and ELR, underscores the importance of AnCon's uncertainty-aware temporal consistency scheme.

**G4:** On rigorous hyperparameter selection procedure and its practical implication
We perform hyperparameter selection without labels instead of looking at the test performance and selecting the best value which is data snooping and can result in overly optimistic performances that are hard to reproduce. Therefore, even though we found (and were not surprised about) that the found hyperparameters were sub-optimal during the sensitivity analysis, we opted to maintain our realistic hyperparameter selection setting.

Another point we should mention is our choice of fixing single hyperparameter values across diverse settings that include 105 numbers of distribution shift scenarios. Of course, even under our aforementioned realistic hyperparameter selection setting, tuning hyperparameters for each scenario could result in better performances. However, we opted to fix the hyperparameter values for all scenarios, considering that in practice the environments (e.g,. Data, distributions, features) change frequently and thus tuning each time is very expensive. By doing so, we believe that we do not report overly optimistic performances of AnCon obtained under settings that are abstracted to the practice. We hope this can resolve Reviewer qtUp’s question on the sub-optimal choice of the hyperparameter value.


**G5:** On performances of baseline methods
In this work, we apply AnCon to two baseline methods, NRC and GCE. While their ideas are simple, these methods are frequently cited as achieving state-of-the-art performances in recent literature (cf. Karim et al., 2023 CVPR; Press et al., 2023 NeurIPS; Rusak et al., 2022 TMLR). The basic intuitions behind these baselines—promoting local consistency and reducing the impact of wrong pseudo labels—are dominant principles in SFDA and TTA literature. To the best of our knowledge, no significantly better methods than these baselines exist without including computationally heavy methods such as ensembles. Therefore, we believe AnCon's compatibility with these baselines demonstrates its compatibility with state-of-the-art methods, and including other similar methods would provide only marginal additional insights.

---

### Decision · Program_Chairs · 2024-09-25

**Decision:**

Accept (poster)

**Comment:**

The paper introduces a novel self-training method (AnCon) to improve test-time adaptation (TTA) and source-free domain adaptation (SFDA) under distribution shifts. It proposes using uncertainty-aware temporal ensembles and label smoothing to enhance pseudo-label quality during self-training. Theoretical guarantees are provided for the method's asymptotic correctness and reduced optimality gap. Extensive experiments demonstrate AnCon's effectiveness in various distribution shift scenarios.

The paper provides a well-founded and efficient approach to improving self-training under distribution shifts, with solid theoretical support and strong empirical results. Although its novelty is incremental, the method's practicality and demonstrated effectiveness across diverse settings make it a valuable contribution to the field. The thorough rebuttals addressed reviewers' concerns. As such, the AC recommends acceptance to NeurIPS.